# A tough and robust hydrogel constructed through carbon dots induced crystallization domains integrated orientation regulation

Huanxin Huo [1,2,4], Jingjie Shen[1,2,4], Jianyong Wan [1,2,3] ✉, Haoran Shi[1,2], Hongxing Yang[1,2], Xin Duan[1,2], Yihong Gao[1,2], Yumeng Chen[1,2], Feng Kuang[1,2], Hongshan Li[1,2], Long Yang [1,2,3] ✉ & Guanben Du [1,2,3] ✉

Tough hydrogels show great potential applied in flexible electronics, sensors and soft robotics, but it remains challenging to combine high strength, toughness and stability. Here, we report the use of carbon dots (CDs) to induce the formation of crystalline domains, to give materials with favourable properties. The CDs act as nanoscale nucleation-sites within polyvinyl alcohol hydrogels, forming dense crystalline domains that serve as physical cross-linking sites. These domains enable a "pinning effect" that enhances energy dissipation and restricts crack propagation. The resulting hydrogels exhibit strong mechanical performance, including tensile strength up to 156 MPa and toughness of 225 MJ m$^{-3}$, while also maintaining good swelling resistance. This strategy is generalizable across different types of CDs and polymer systems. In addition, the hydrogels demonstrate stable conductivity under water, making them suitable for applications in underwater motion sensing and flexible supercapacitors. This work provides a scalable approach to engineer robust, multifunctional hydrogels.

Hydrogels are extensively used in fields of wearable electronics[1], soft robotics[2], nano-friction generators[3], biomedicine[4], and solid-state electrolytes[5], etc. due to their unique electrical conductivity, flexibility and three-dimensional networks. However, conventional hydrogels consist of a single hydrophilic polymer network and lack an energy dissipation mechanism, which makes them susceptible to stress concentration and results in weak tensile strength and toughness. Achieving a balance between strength and toughness in hydrogels is inherently challenging, making it difficult to attain both high toughness and strength simultaneously. To address these challenges, researchers have developed hydrogels with enhanced strength and toughness through various strategies, including the construction of dual networks[6], organic-inorganic hybridization[7], densification[8], synergistic interactions[9], directional freezing[10], and salt precipitation[11], resulting in significant improvements in mechanical properties.

Furthermore, the toughness of hydrogels can be enhanced by regulating the motion of macromolecular chains. Wu et al.[12] found that crystalline domains slow down crack propagation due to the "pinning effect" during stretching, thereby enhancing the toughness of the hydrogels. Despite these advancements, it remains challenging to achieve both high fracture strength and toughness in the material.

Traditional nanofillers (such as carbon nanotubes, graphene, and MXenes, etc.) have garnered significant attention due to their interfacial effects with polymers[11]. Wang et al.[13] prepared hydrogels with high mechanical properties (tensile strength of 9.4 MPa and strain of 360%, respectively) using aramid nanofibers. However, the poor compatibility of nanofillers with the matrix often results in poor dispersion, undermining the effectiveness of the reinforcement strategy. Such interfacial interactions are typically based on weak physical interactions and limited covalent bonding, leading to only marginal

[1]Yunnan Province Key Lab of Wood Adhesives and Glued Products, Southwest Forestry University, Kunming, China. [2]College of Materials and Chemical Engineering, Southwest Forestry University, Kunming, China. [3]International Joint Research Center for Biomass Materials, Southwest Forestry University, Kunming, China. [4]These authors contributed equally: Huanxin Huo, Jingjie Shen. ✉e-mail: jywan@swfu.edu.cn; lyang@swfu.edu.cn; guanben@swfu.edu.cn

improvements in mechanical properties of hydrogels[14]. Additionally, traditional nanofillers are expensive and cumbersome to produce, which lead to the gradual emergence of carbon dots (CDs) as an alternative. As a new member of the "carbon family", CDs are carbon nanoparticles with sizes <10 nm, which were discovered accidentally by Xu et al.[15] in 2004 during the preparation of electrophoretically purified single-walled carbon nanotubes. In 2006, Sun et al.[16] prepared luminescent CDs through surface passivation. Since then, this field has attracted numerous researchers due to its excellent optical properties[17], dispersibility[18], low toxicity[19], and high biocompatibility[20]. Notably, as a member of the carbon family, CDs not only possess attractive $sp^2$ nuclei that provide prominent interfacial interactions (e.g., π–π stacking, CH–π interactions, etc.) but also retain the functional groups of carbonized precursors after carbonization, exhibiting good dispersibility. Consequently, Wan et al.[21] introduced CDs and then prepared robust composite with a strength of 124 MPa.

Herein, we proposed a strategy involving CDs-induced crystalline domains and the construction of an ordered structure based on "pinning effect" to prepare robust hydrogels. The generation of numerous crystalline domains in poly(vinyl alcohol) (PVA) was induced using citric acid CDs, importantly, the addition of CDs significantly altered the modulus and toughness of hydrogel, changing it from 0.06 MPa and 1.51 MJ m$^{-3}$ (PVA) to 3.9 MPa and 33.99 MJ m$^{-3}$ (PVA–CDs), severally. Subsequently, the PVA–CDs hydrogel was used to construct a strong hydrogel (PVA–CDs–SP) with an ordered structure based on the "pinning effect", it was achieved in conjunction with the introduction of sodium citrate (Na$_3$Cit), enhanced the environmental tolerance of hydrogel by inhibiting the formation of hydrogen bonds between H$_2$O molecules through ionic hydration with water molecules. As-prepared PVA–CDs–SP hydrogel exhibited high strength (156 MPa) and toughness (225.2 MJ m$^{-3}$), far exceeding those of elastomers, engineering plastics, rubbers, bionic tendons and artificial spider silk, etc. Furthermore, the PVA–CDs–SP hydrogel could support a weight ~1.5 × 10$^5$ times its own. This strategy is versatile and directly applicable to various combinations of Hofmeister effect-sensitive polymers and CDs, offering an approach for fabricating hydrogels with high mechanical properties.

## Results

Due to their nanostructures and excellent interfacial interactions, CDs can enhance the mechanical properties of hydrogels. Additionally, the mechanical properties of hydrogels are highly correlated with the degree of densification of the polymer chains and crystalline domains[22]. Therefore, the CDs was introduced to generate larges number of crystalline domains, and then combined with the orientation strategy based on "pinning effect" to prepare robust hydrogels.

Citric acid (CA) with abundant carboxyl groups and hydroxy groups, was used to prepare CDs (Fig. 1a). The transmission electron microscope (TEM) image of CDs revealed that it possessed a quasi-spherical morphology, and the lattice structure with a spacing of 0.19 nm was clearly visible (Fig. 1b and Supplementary Fig. 1). Additionally, the particle size distribution of the CDs exhibited a prominent peak centered around ~3 nm (Supplementary Fig. 2a), with a PDI (polydispersity index) of 0.159, indicating a relatively narrow and uniform size distribution. Furthermore, the zeta potential analysis revealed that the CDs surface carried a positive charge, with a zeta potential of ~+13 mV (Supplementary Fig. 2b), suggesting that the CDs possessed good dispersibility and stability, preventing particle aggregation or precipitation in solution. The relatively high zeta potential further confirmed the excellent stability of the sample.

In Raman spectra, the peak in the D band at 1340 cm$^{-1}$ was associated with disorder structure, while the peak in the G band at 1574 cm$^{-1}$ corresponded to the vibrations of $sp^2$-hybridized structure. In this experiment, the ratio of the intensity of G band and D band was 1.03, indicating a high degree of graphitization in the CDs (Fig. 1c). The

particle sizes of the CDs that their diameters range from 2 to 5 nm, with an average size of 3.5 nm. CDs exhibited a distinct broad diffraction peak at 28.4° (Fig. 1d). The –OH and –COOH peaks corresponding to CDs were observed at 3289 and 1755 cm$^{-1}$, respectively. Besides, a peak for C = C at 1630 cm$^{-1}$ was observed, which illustrated the transformation from C–C to C = C after carbonization (Supplementary Fig. 3). It displayed the high-resolution energy spectrum in C1s scan, revealing peaks at 284.8 eV for C–C/C = C, 285.7 eV for C–O, 286.5 eV for C = O, and 289.2 eV for O–C = O and π–π*. In O1s scan, it showed three peaks at 532.6, 533.3, and 534.2 eV, corresponding to C = O, O–H, and C–O, respectively (Supplementary Fig. 4). These phenomena indicated that the original functional groups of citric acid precursor were retained after carbonization[21].

As shown in Fig. 1e, the PVA hydrogel was prepared and then the PVA–CDs hydrogel was obtained by the introduction of CDs with $sp^2$ nuclei to the PVA hydrogel to enhance its crystallinity. Salt precipitation in sodium citrate solution was then performed to promote the formation of new crystalline domains in the hydrogel, which increased the non-covalent crosslinking density of the PVA networks. As-prepared hydrogel was subsequently strengthened through energy dissipation between the polymer chains during stretching, and the "pinning effect" also enhanced the stability of the PVA networks. The increased crystalline domains from salt precipitation slowed crack propagation and aligned the disordered regions with the crystalline areas of the macromolecular chains, reducing the distance between the chains and the crystal spacing. This densification elevated the fracture strength and toughness of the hydrogel, as the PVA chains, influenced by the "pinning effect", did not revert to disordered long chains, further improving its mechanical properties. The scanning electronic microscopy (SEM) images of the surface and cross-section of the freeze-dried hydrogel showed that the PVA surface was rough, with small pore structures on the cross-section. In contrast, the surface of PVA–CDs was smooth after the addition of CDs. The PVA–CDs–SP hydrogel, constructed based on the "pinning effect", exhibited a pronounced oriented structure on the surface and an abundance of nanofibers in the cross-section, which just like a tendon (Fig. 1f and Supplementary Fig. 5).

## Structure evolution and toughening mechanism of hydrogels

To elucidate the contribution of CDs to the induced crystallization domains in hydrogels, the PVA, PVA–CA, and PVA–CDs hydrogels were prepared. With the addition of CDs (Fig. 2a), the fracture stress of the hydrogels increased from 0.27 to 9.36 MPa. This enhancement was attributed to the interactions between the CDs and PVA chains (e.g., hydrogen bonding and van der Waals forces, etc.), which promoted the aggregation of the PVA chains, resulting in enhanced mechanical properties for the PVA–CDs hydrogels. Notably, the strength and toughness were optimally enhanced with the addition of 10 wt% CDs, due to the typical nano-enhancement effect of CDs (Supplementary Fig. 6). The abundant hydroxyl and carboxyl groups on the surface of CDs provided additional cross-linking sites and nucleation sites, leading to more stable and denser networks. Additionally, the nanoscale size and good dispersion of CDs enhanced the density of hydrogels networks, which could effectively withstand external loads, thereby increasing tensile strength and toughness. For hydrogels with different PVA concentrations, the addition of PVA also conferred excellent mechanical properties (Supplementary Fig. 7).

To further investigate the effect of CDs for mechanical properties of hydrogel, the hydrogen bonding networks of neighboring water molecules were studied using Fourier-transform infrared (FT-IR) and Raman spectroscopy. The PVA, PVA–CA, and PVA–CDs hydrogels exhibited a pronounced O–H vibrational peak near 3248 cm$^{-1}$ (Fig. 2c). For the PVA–CA hydrogel, the addition of CA altered the vibration of O–H compared to pure PVA hydrogels, indicating the formation of hydrogen bonds between PVA and CA. Notably, the peak in PVA-CDs

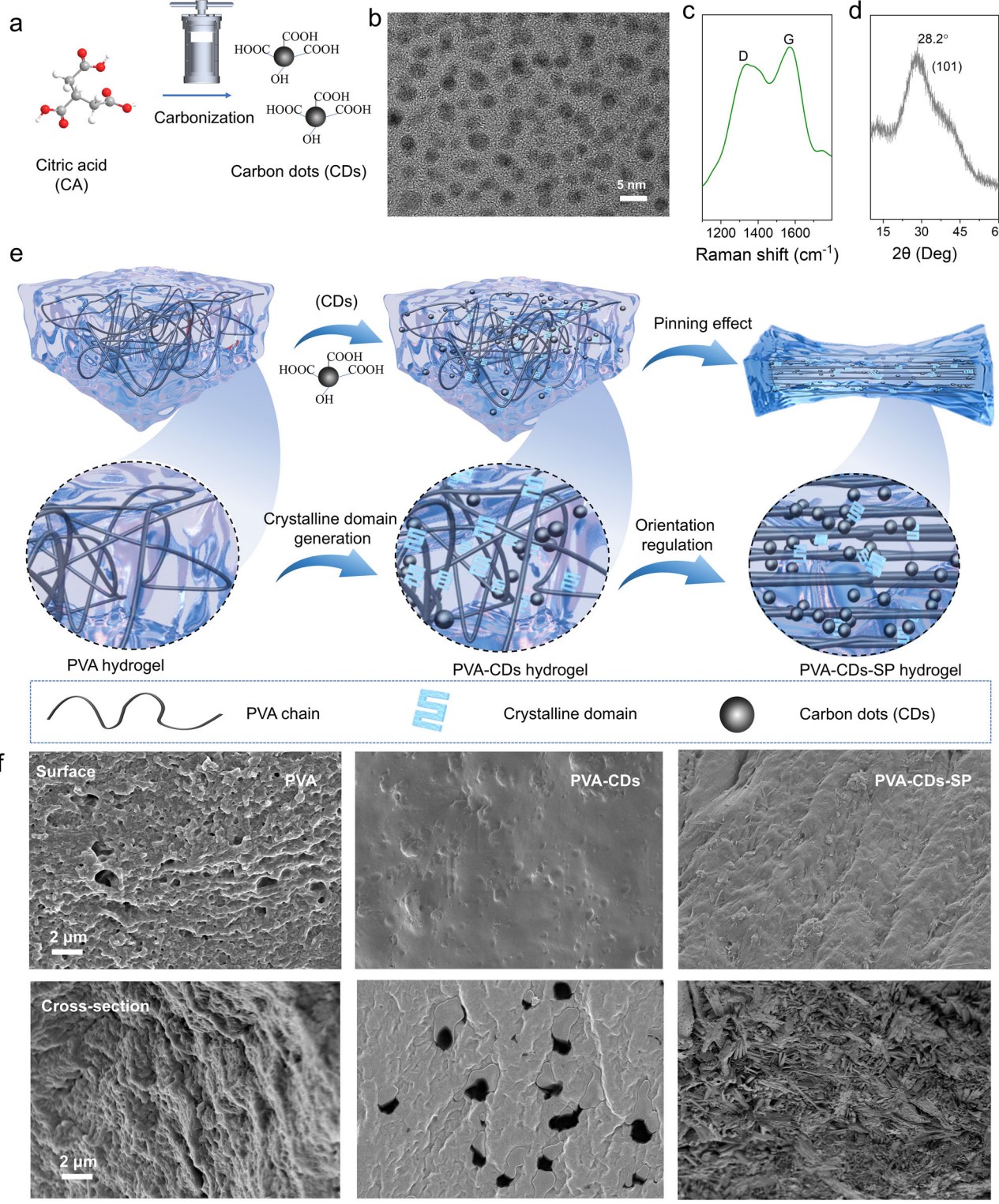

**Fig. 1 | Preparation of CDs and PVA–CDs–SP hydrogel. a** Schematic diagram of the preparation of CDs. **b** Transmission electron microscopy (TEM) images of CDs and its particle size distribution (inset). **c** Raman spectrum of CDs. **d** X-ray diffraction (XRD) spectrum of CDs. **e** Schematic diagram of the preparation of PVA–CDs–SP hydrogel. **f** Scanning electron microscopy (SEM) images of the surfaces (up) and cross-sections (down) of PVA, PVA–CDs, and PVA–CDs–SP hydrogel.

hydrogel shifted from 3248 to 3240 cm$^{-1}$ with the addition of CDs, suggesting that the interaction between CDs and polymer was enhanced. As shown in Fig. 2d, the peaks in the range of 3000–3800 cm$^{-1}$ corresponded to the O–H stretching vibrations. The peaks near 3200 cm$^{-1}$ were associated with water molecules bound by strong hydrogen bonding, while the peak around 3400 cm$^{-1}$ corresponded to bound water via weak hydrogen bonding[23]. The addition of CA showed little effect on the hydrogen bonding networks of the PVA–CA hydrogel. The introduction of CDs led to a significant increase in the ratio of strong to weak hydrogen bonding ($I_{strong}/I_{weak}$), indicating that CDs enhanced the strong hydrogen bonding interactions between water molecules. Similarly, the –OH peak in PVA–CDs

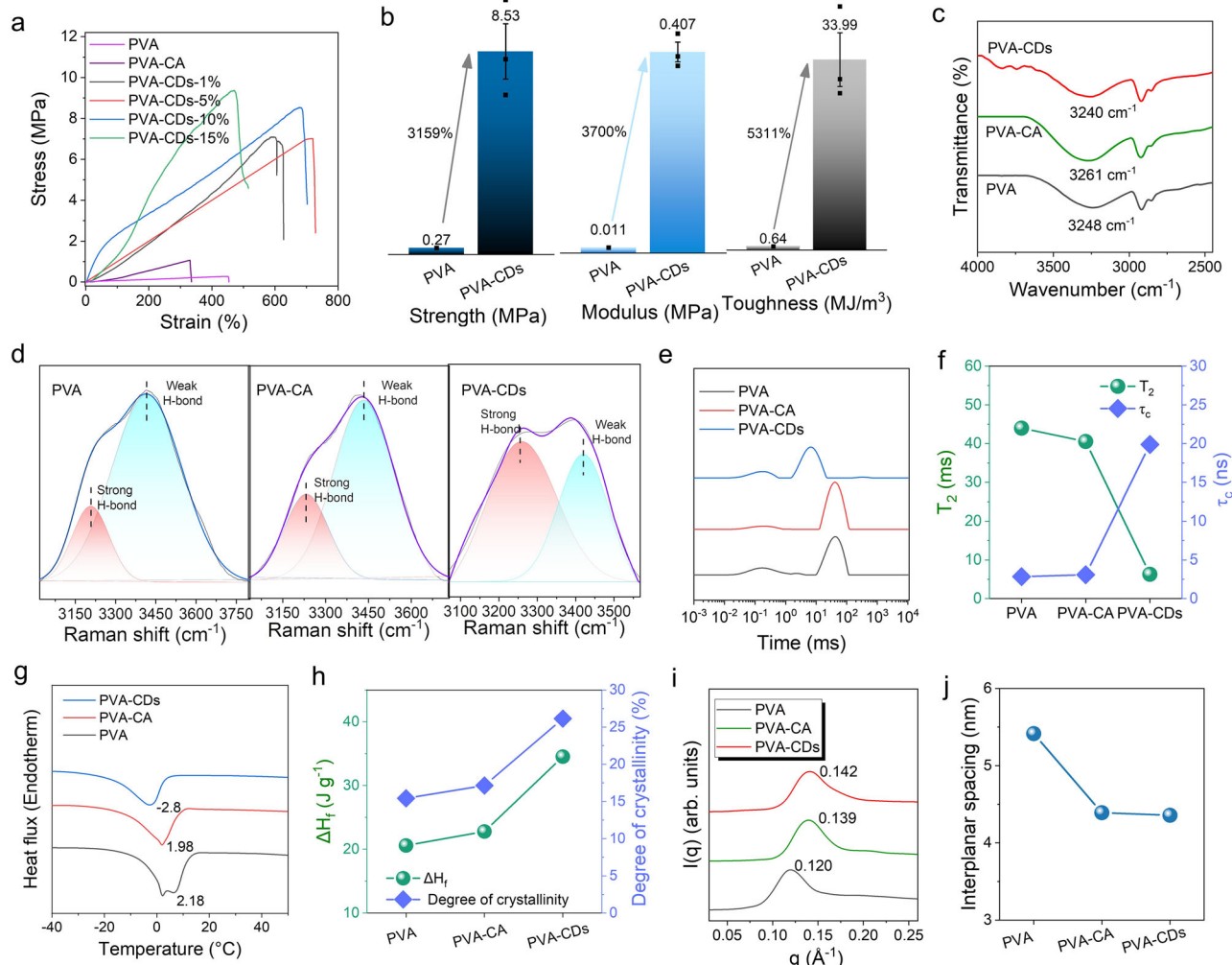

**Fig. 2 | Mechanical properties of hydrogels and toughening tuning via the addition of CDs. a** Stress–strain curves of PVA–CDs hydrogels at different concentrations of CDs. **b** Comparison of the mechanical properties of PVA, PVA–CA, and PVA–CDs hydrogels. The bars represent the average values of three independent measurements, with error bars indicating the standard deviation. **c** FT-IR spectra of PVA, PVA–CA, and PVA–CDs hydrogels. **d** Raman spectra of the O–H stretching vibrational peaks of PVA, PVA–CA, and PVA–CDs hydrogels. **e** The low-field nuclear magnetic resonance (L-NMR) curves of PVA, PVA–CA, and PVA–CDs hydrogels. **f** The corresponding $T_2$ and $\tau_c$ values of PVA, PVA–CA, and PVA–CDs hydrogels. **g** DSC analysis of PVA, PVA–CA, and PVA–CDs hydrogels. **h** Enthalpy changes of melting and crystallinity of PVA, PVA–CA, and PVA–CDs hydrogels. **i** SAXS spectra of PVA, PVA–CA, and PVA–CDs hydrogels. **j** The interlayer spacing of PVA, PVA–CA, and PVA–CDs hydrogels.

shifted from 532.2 eV to a lower energy region compared to PVA hydrogels in (X-ray photoelectron spectroscopy) XPS, and parallel phenomena were observed in C1s scan, indicating that the introduction of CDs elevated hydrogen bonding interactions (Supplementary Figs. 8, 9). To verify the increase in bound water in hydrogels, low-field nuclear magnetic resonance (L-NMR) tests were employed to confirm the highly bound state of water in these hydrogels (Fig. 2e). The proton spin-spin relaxation time ($T_2$) of water in PVA–CDs hydrogels was 6.29 ms, significantly shorter than that of PVA (43.98 ms) and PVA–CA (40.55 ms) (Fig. 2f). Besides, the $\tau_c$ of PVA–CDs (19.89 ns) was significantly larger than that of PVA (2.84 ns) and PVA–CA (3.08 ns), suggesting a decrease in the mobility of water molecules. This could be attributed to the enhanced strong hydrogen bonding restricting the mobility, resulting in an increase in bound water and a decrease in the mobility of remaining free water.

To investigate the effect of CDs incorporation on the crystallinity of the hydrogel, XRD characterization of PVA–CDs prepared at different CDs concentrations was performed, as shown in Figure (Supplementary Fig. 10). As the CDs concentration increases, the peak at 19.5° corresponding to the (101) crystal plane gradually increased,

indicating that the introduction of CDs significantly enhanced the crystallinity of the hydrogel. The introduction of CDs decreased the freezing point (Fig. 2g), which was attributed to increased internal dispersion, heterogeneity, and bound water in hydrogels, leading to a decrease of crystallization. The typical (101) reflection of hydrogel for PVA, PVA–CA, and PVA–CDs appeared at 19.5° (Supplementary Fig. 11). The intensity of the peak for PVA–CDs hydrogel was significantly higher than that for PVA and PVA–CA hydrogels, which confirmed the crystallinity of hydrogels increased due to the introduction of CDs, where, the crystallinity of PVA-CDs hydrogel was 26.1%, which was much higher than PVA (15.4%) and PVA-CA hydrogels (17.1%) (Supplementary Fig. 12 and Fig. 2h). The addition of CDs significantly enhanced the crystallinity of the hydrogel, primarily due to their role as heterogeneous nucleating agents. CDs provided effective nucleation sites, reducing the required undercooling for crystallization and promoting the orderly arrangement of polymer chains within the hydrogel. The surfaces of the CDs were rich in functional groups, which could interact with polymer chains or ions, further facilitating the formation and growth of crystallization nuclei. This increased the number and size of crystalline regions, thereby improving the overall crystallinity.

Furthermore, the presence of CDs could strengthen the interactions between the crystalline and amorphous regions, enhancing the mechanical properties of hydrogel. Thus, the incorporation of CDs not only enhanced the crystallinity of the hydrogel but also optimized its structure, leading to improved its mechanical performance.

Small-angle X-ray scattering (SAXS) was employed to investigate the characteristics of the crystalline domains in PVA–CDs hydrogels. The 2D-SAXS diffractograms of the hydrogels showed the random distribution of crystalline domains (Supplementary Fig. 13). The SAXS spectra (Fig. 2i) exhibited scattering peaks at 0.120 Å for PVA, 0.139 Å for PVA–CA and 0.142 Å for PVA–CDs hydrogels. According to Bragg's equation, the average distances ($L$) between adjacent crystalline domains in PVA, PVA–CA, and PVA–CDs hydrogels were 5.41, 4.39 and 4.35 nm, respectively (Fig. 2j), which indicated that the increase of crystallinity was due to the increase in the number of crystalline domains rather than the formation of larger crystalline domains.

## Extensive tuning of the mechanical properties of hydrogels induced by "pinning effect"

The oriented hydrogel (PVA–CDs–SP) was constructed based on "pinning effect". The results indicated that the optimal temperature for preparing CDs was 160 °C. At this temperature, the tensile strength of the PVA–CDs–SP hydrogel prepared from the CDs reached 154.5 MPa, which was significantly higher than that of hydrogels prepared from CDs carbonized at other temperatures (Fig. 3a). In addition, the toughness of the PVA–CDs–SP hydrogel prepared from the CDs carbonized at 160 °C reached its maximum value of 210.3 ± 21.9 MJ m$^{-3}$ (Fig. 3b). A comparison of the fracture stresses of PVA, PVA–CDs, PVA–CDs–S, and PVA–CDs–SP hydrogels revealed that the orientation structure induced by "pinning effect" significantly enhanced the mechanical properties of hydrogel (Fig. 3c). Specifically, the tensile strength, modulus and toughness of PVA–CDs–SP compared to PVA hydrogel increased by 53,807%, 514,581%, and 32,853%, respectively (Fig. 3d).

Additionally, different biomass (such as malic acid, tannic acid and eucalyptus bark, etc.) was applied to prepared CDs, and further fabricated the PVA–CDs–SP hydrogels to investigate the generalizability (Fig. 3e, f). As anticipated, the PVA–CDs–SP hydrogels constructed with different CDs carbonized from malic acid, tannic acid, and eucalyptus bark similarly exhibited excellent fracture stresses (129.37, 52.23, and 85.85 MPa, respectively) and toughness values (202.8, 86.5, and 148.6 MJ m$^{-3}$, respectively), indicating that our strategy is universally applicable to different biomass-based CDs. During this process, the structure of the hydrogel gradually transformed from an initial disordered state to an ordered structure, resulting in significant improvements in both tensile strength and toughness. Additionally, to illustrate the changes in the mechanical properties of the hydrogel during the progressive stretching process, PVA–CDs–SP hydrogels under different progressive stretches were tested, as shown in Supplementary Fig. 14a. As the progressive stretching proceeded, the mechanical properties of the PVA–CDs–SP hydrogel gradually increased, but this was accompanied by a decrease in the elongation at break, which was due to the fact that after extensive progressive stretching, the PVA–CDs–SP hydrogel gradually approached an oriented state. Furthermore, as the progressive stretching continued, the water content of the PVA-CDs-SP hydrogel gradually decreased (Supplementary Fig. 14b), because during the stretching process, the PVA chains tended to align, causing the internal water content to decrease. To further confirm the ability of PVA–CDs–SP hydrogel to retard crack propagation, the crack sensitivity of the PVA–CDs–SP hydrogel was also investigated. Interestingly, we observed that with the increase in pre-stretching, the arrangement of polymer chains in the hydrogel became significantly enhanced, leading to a higher degree of orientation. However, this also resulted in a significant reduction in the

elongation ability of the notched hydrogel (Supplementary Fig. 15a). Therefore, although the orientation was improved, the measured fracture energy did not increase as expected. This suggests that, while the oriented structure was beneficial for tensile strength and stiffness, in a highly ordered state, it might simultaneously limit energy dissipation around the crack tip (Supplementary Fig. 15b). To investigate the mechanical anisotropy of the material, the tensile strength of the hydrogel in the vertical direction was measured (Supplementary Fig. 16). Clearly, in the vertical direction, the fracture strength and fracture elongation of PVA–CDs–SP hydrogel in the oriented direction were significantly higher than the corresponding values in the vertical direction, which was a result of the molecular alignment in the oriented direction.

The PVA–CDs–SP hydrogel showed higher tensile strength and toughness compared to other reported hydrogels. (Fig. 3g)[24–41]. Furthermore, the toughness of PVA–CDs–SP significantly surpasses that of anhydrous polymers such as polydimethylsiloxane (PDMS), Kevlar, and synthetic rubber, even exceeding the toughness of natural tendon and spider silk (Fig. 3h)[30], suggesting the potential for replacing these materials in various applications. The construction of the oriented structure significantly increased the Young's modulus of the PVA–CDs–SP hydrogel, elevating it from the original gel level to that of rubber (Fig. 3i). In addition, the hydrogel is capable of lifting ~30 kg, which is roughly $1.5 \times 10^5$ times its own weight. To further demonstrate the mechanical properties of the PVA–CDs–SP hydrogel, we arranged the PVA–CDs–SP hydrogel into multiple parallel strands, then bound the two ends of the strands together with several cable ties to form a rope. This rope was capable of supporting the weight of a 90 kg adult and even pulling a car (~$2.0 \times 10^3$ kg) (Fig. 3j–l, and Supplementary movies 1, 2).

## Toughening mechanism of hydrogels

In the FT-IR spectra, the characteristic peak of –OH in hydrogel gradually shifted from 3240 cm$^{-1}$ (PVA–CDs) to 3217 cm$^{-1}$ (PVA–CDs–SP) with the formation of the oriented structure, indicating that this synergistic effect enhanced the hydrogen bonding interactions in hydrogels (Fig. 4a). The peak of –OH gradually shifted from 532.1 to 531.9 eV, and it also revealed parallel phenomena in C1s scan, confirming that the formation of the ordered structure enhanced the hydrogen bonding interactions in hydrogels in the XPS spectra (Supplementary Figs. 17, 18). Meanwhile, the Raman spectrum displayed a significant enhancement in the ratio of $I_{strong}/I_{weak}$ (Fig. 4b). Low-field nuclear magnetic resonance (L-NMR) tests confirmed that the $T_2$ of water in the hydrogel decreased significantly during this treatment (Fig. 4c, d), while $\tau_c$ exhibited a significant increase, indicating an increase in bound water and a decrease in the mobility of the remaining free water. These results indicated that the formation of ordered structures enhanced the strong hydrogen bonding interactions in hydrogels.

The process of forming ordered structures also lowered the freezing point to certain extent (Fig. 4e). The hydrogels likely underwent structural rearrangement, resulting in more compact water molecules, which in turn led to a significant decrease in the freezing point. Subsequently, XRD was used to analyze the evolution of the crystalline domains during the salt-assisted progressive stretching process. The peak at 19.5° gradually increased (Supplementary Fig. 19), indicating that the crystallinity gradually increased. The crystallinity of PVA-CDs, PVA-CDs-S, and PVA-CDs-SP were calculated to be 26.14%, 36.74%, and 49.05%, respectively, based on the DSC results (Fig. 4f and Supplementary Fig. 20). This was because when the hydrogel was subjected to strain, the polymer chains were stretched and aligned along the direction of the applied force. This molecular alignment promoted the nucleation of crystallites and their growth in a more ordered and directional manner, thereby enhancing the crystallinity of the material. Strain facilitated the rearrangement of amorphous

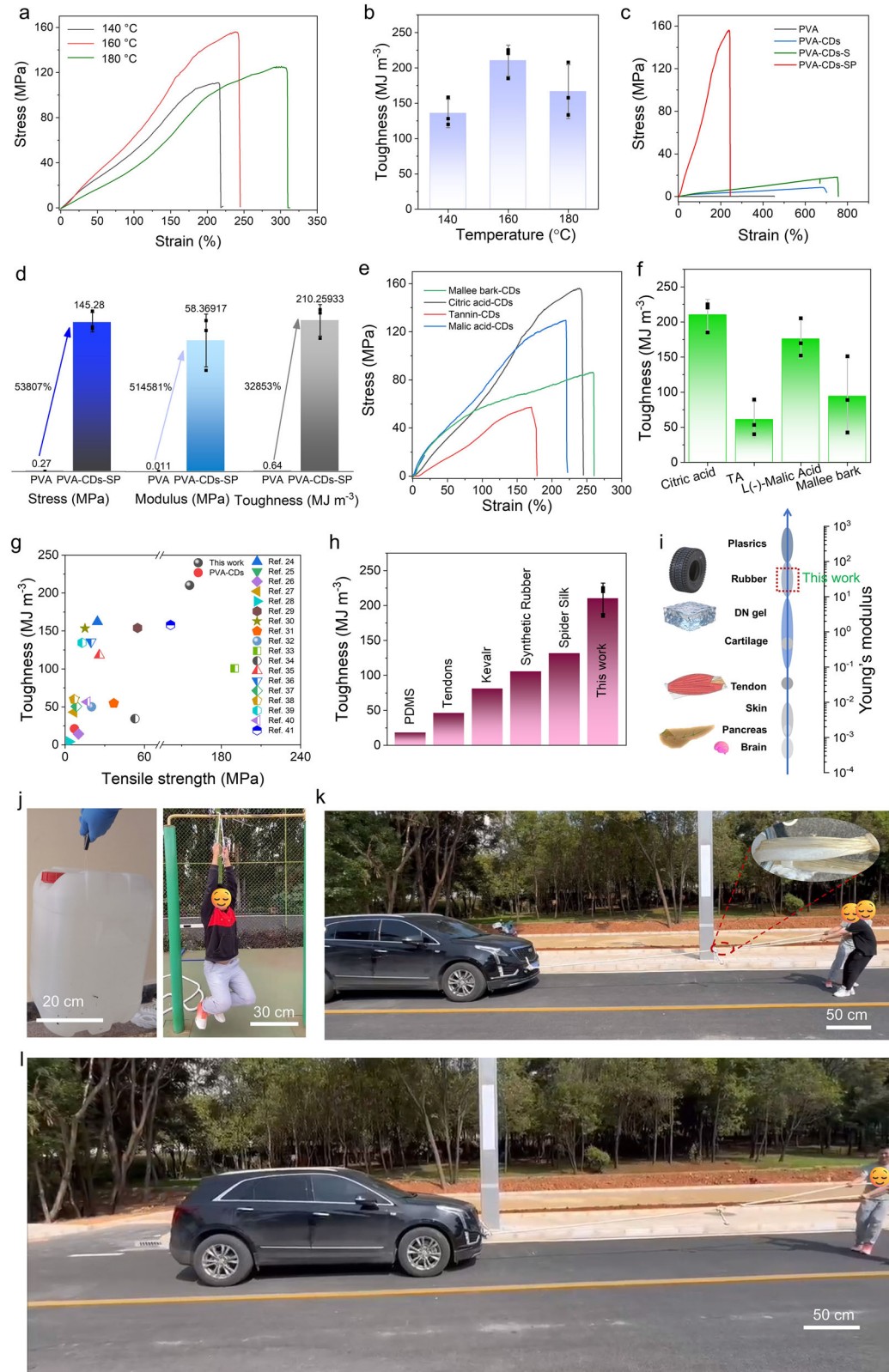

polymer segments, encouraging them to adopt a more organized structure. As the strain increases, the alignment of polymer chains became more pronounced, and the size and number of crystallization domains increased.

The applied strain also affects the shape and distribution of the crystalline regions, making them more anisotropic. The directional growth of these crystallites under mechanical stress led to an overall

increase in the structural integrity of the hydrogel. Stretching the hydrogel could reduce the energetic barriers to crystallization, allowing crystallites to nucleate more uniformly and rapidly, further supporting the crystallization process. The higher the applied strain, the more aligned and densely packed the crystalline domains become, resulting in a stronger hydrogel with enhanced mechanical properties such as tensile strength, stiffness and toughness.

**Fig. 3 | Mechanical properties of tough and robust hydrogels induced by "pinning effect". a** Stress–strain curves of PVA–CDs–SP hydrogels prepared with CDs at different carbonization times. **b** Toughness of PVA–CDs–SP hydrogels prepared with CDs at different carbonization times. The bars represent the average values of three independent measurements, with error bars indicating the standard deviation. **c** Stress–strain curves of PVA, PVA–CDs, PVA–CDs–S and PVA–CDs–SP hydrogels. **d** Comparison of the mechanical properties between PVA and PVA–CDs–SP hydrogels. The bars represent the average values of three independent measurements, with error bars indicating the standard deviation. **e** Stress–strain curves of PVA–CDs–SP hydrogels prepared with CDs derived from various biomass sources. **f** Toughness of PVA–CDs–SP hydrogels prepared with CDs derived from various biomass sources. The bars represent the average values of three independent measurements, with error bars indicating the standard deviation. **g** Comparison of the fracture stress and toughness of PVA–CDs–SP hydrogels with other reported hydrogels. **h** Comparison of the mechanical properties of PVA–CDs–SP hydrogel with other high-toughness materials. The bars represent the average values of three independent measurements, with error bars indicating the standard deviation. **i** Modulus range of PVA–CDs–SP hydrogels. **j** PVA–CDs–SP hydrogels lifted heavy weights. **k** and **l** The PVA–CDs–SP hydrogel pulled a car.

Furthermore, strain-induced crystallization works synergistically with the "pinning" effect, where the crystalline regions acted as physical barriers to macroscopic deformation. The presence of these pinning points restricted the mobility of the amorphous chains, thereby enhancing the ability of hydrogel to withstand larger deformations without catastrophic failure. Therefore, strain not only influenced the size and orientation of the crystallization domains but also improved the overall mechanical properties of the hydrogel by reinforcing its internal structure.

The anisotropic structures of hydrogels were further investigated using small-angle X-ray scattering (SAXS). The regular diffraction rings of PVA-CDs-S and PVA-CDs-SP hydrogels transformed into arc-shaped rings, clearly, indicating the presence of an oriented structure within the hydrogel (Fig. 4g). With the implementation of this strategy based on "pinning effect", the intensity at the azimuthal angle of 90° gradually increased. The Herman orientation parameters ($f_c$) of PVA-CDs, PVA-CDs-S and PVA-CDs-SP, determined from the azimuthal integration intensity distribution curves, were 0.333, 0.783 and 0.794, respectively (Fig. 4h). The high $f_c$ values further confirmed the high degree of orientation of the hydrogels. Additionally, the SAXS spectra of the hydrogels (Fig. 4i) revealed that PVA-CDs hydrogel exhibited a scattering peak at 0.142 Å, while the scattering peaks of PVA-CDs-S and PVA-CDs-SP were observed at 0.159 and 0.19 Å, respectively. According to Bragg's equation, the average distances between neighboring crystalline domains of PVA-CDs, PVA-CDs-S, and PVA-CDs-SP hydrogels were ~4.36, ~4.0, and ~3.40 nm (Supplementary Fig. 21), respectively, which indicated that our strategy based on "pinning effect" increased the number of crystalline domains, and it was distinct that the increase of the number of crystalline domains.

It was unambiguous that salting out and progressive stretching were beneficial for increasing the crystallinity of PVA hydrogels, due to salting out could induce PVA aggregation and crystallization by phase separation. Besides, during progressive stretching, the polymer chains were elongated and tend to align. This orientation led to stronger interchain interactions, promoting the formation of crystals. Furthermore, the crystallinity that developed during salt precipitation-assisted progressive stretching enhanced the mechanical properties of the hydrogel, as the crystalline domains acted as rigid, high-functionality cross-linkers.

To verify the formation of crystalline domains in PVA-CDs-SP hydrogels during progressive stretching, XRD and DSC tests were conducted. As shown in Supplementary Fig. 22a, with progressive stretching, the peak at 19.5° gradually increased, indicating an increase in the crystallinity of the PVA-CDs-SP hydrogel. The DSC test also demonstrated an increase in the crystallinity of the PVA-CDs-SP hydrogel (Supplementary Fig. 22b, c), with the enthalpy change gradually increasing between 200 and 250 °C. These results indicated that the crystallinity of the PVA-CDs-SP hydrogel significantly increased during progressive stretching. To confirm that the increase in crystallinity of the PVA-CDs-SP hydrogel was not due to the formation of large crystalline domains, SAXS testing was performed. As shown in Supplementary Fig. 23a, with progressive stretching, the 2D-SAXS image gradually transformed into sharp rings, indicating an increase in orientation. Additionally, the SAXS spectrum showed that with the

degree of stretching, the value of $q_{max}$ gradually increased (Supplementary Fig. 23b), with a significant shift. This suggested a clear reduction in the interplanar spacing of the PVA-CDs-SP hydrogel (Supplementary Fig. 23c), further proving that the increase in crystallinity of the PVA-CDs-SP hydrogel was not due to the formation of large crystalline domains but rather the growth of more small crystalline domains.

The polarized light microscope images showed that the sample exhibited significant changes in its polarized light response intensity under different stretching magnifications (100%, 200%, 300%, 400%, 500%, 600%) and different rotation angles (0°, 45°, 90°), which proved the orientation of crystallization during the stretching process (Supplementary Fig. 24). As the stretching magnification increases, especially above 400%, the brightness of the polarized images significantly increased, and distinct streak-like or fibrous bright regions appeared in the images, indicating that the crystalline regions tended to align along the stretching direction, and optical anisotropy gradually strengthened. At low stretching magnifications (100%, 200%), the image differences under different rotation angles (0°, 45°, 90°) are small, suggesting that the crystals had not significantly oriented. However, as the stretching magnification increases, particularly in the range of 400–600%, the brightness of the images changed noticeably at different angles, showing clear anisotropy. In particular, at a 45° angle, the polarization response is the strongest, indicating a significant interaction between the crystal alignment direction and the polarization direction.

This enhanced brightness contrast with increasing stretching suggested that the crystallites underwent oriented growth during the stretching process and exhibited a pronounced birefringence effect. Overall, the results confirmed that the PVA induced by CDs underwent oriented crystallization under stretching, and at high stretching magnifications, the alignment of chain segments along the stretching direction was enhanced, thereby promoting the oriented growth of the crystals. This directional crystallization effect also reflected the fact that CDs and strain enhanced the binding effect of the hydrogel through a synergistic interaction. CDs as heterogeneous nucleating agents, promoted the formation of crystalline regions and provided additional pinning points, thereby restricting the free sliding of polymer chains in the amorphous regions. With the application of strain, the polymer chains rearranged and promoted the directional growth of the crystalline regions, further enhancing the pinning effect. Strain increased the number and density of crystalline domains, enabling the hydrogel to resisted deformation and fracture more effectively during the deformation process. The combined effect of CDs and strain enhanced the mechanical properties of hydrogels, such as tensile, compressive, and fatigue resistance, by increasing crystallinity and creating dense binding points, thereby significantly improving their structural stability and mechanical performance.

## Application for underwater sensing

To assess the stability of PVA–CDs–SP hydrogels, 1100 loading-unloading cycles were performed at a strain of 30% (Fig. 5a). The load-unload curves showed good overlap, indicating low mechanical hysteresis and excellent recovery of the hydrogel. In the first hundred

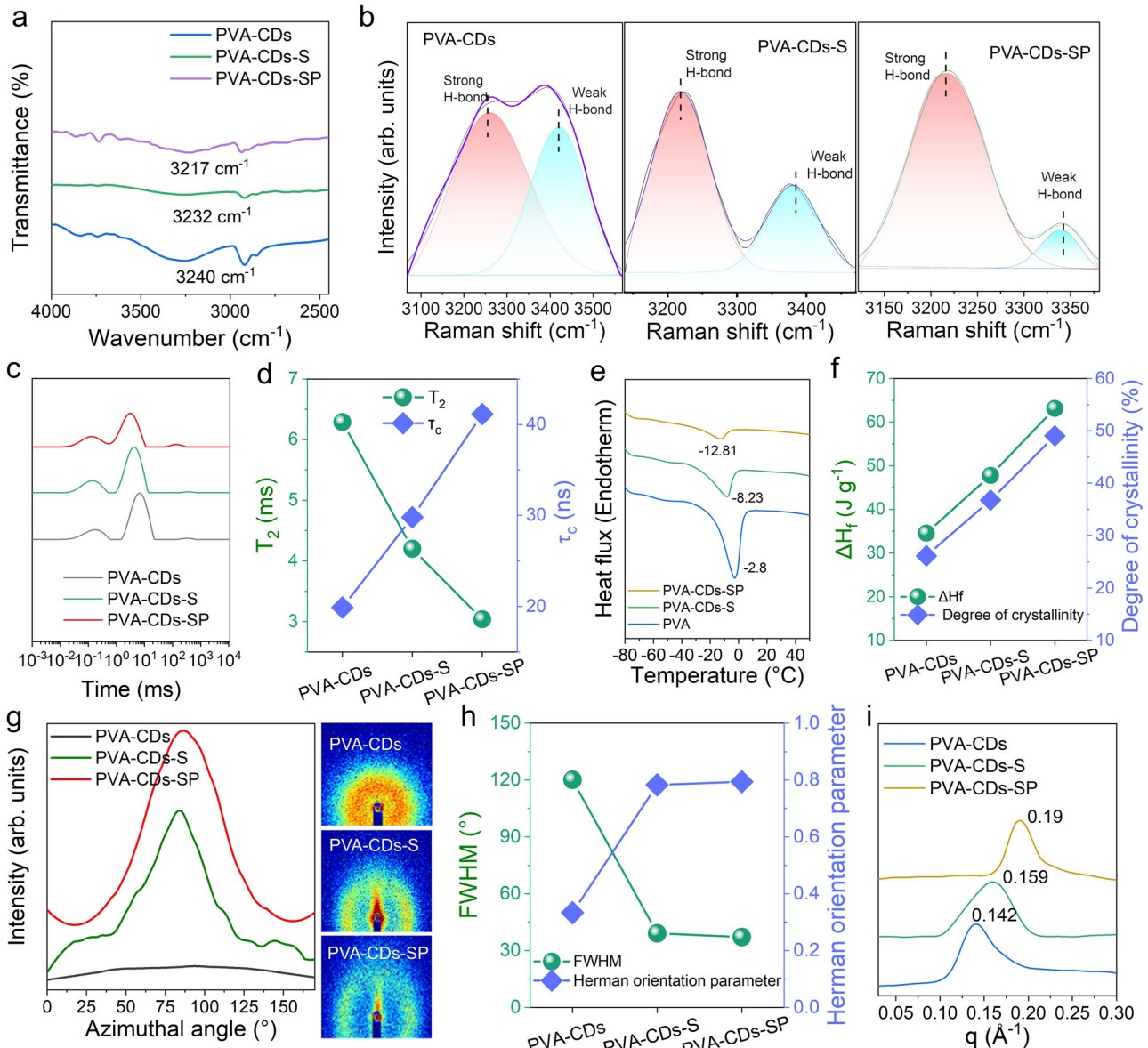

**Fig. 4 | Toughening mechanism of hydrogels. a** FT-IR spectra of PVA–CDs, PVA–CDs–S, and PVA–CDs–SP hydrogels. **b** Raman spectra of PVA–CDs, PVA–CDs–S, and PVA–CDs–SP hydrogels. **c** Low-field nuclear magnetic resonance (L-NMR) curves of PVA–CDs, PVA–CDs–S, and PVA–CDs–SP hydrogels. **d** Corresponding $T_2$ and $\tau_c$ values for PVA–CDs, PVA–CDs–S and PVA–CDs–SP hydrogels. **e** DSC analysis of PVA–CDs, PVA–CDs–S, and PVA–CDs–SP hydrogels.

**f** Melting enthalpy and crystallinity of PVA–CDs, PVA–CDs–S, and PVA–CDs–SP hydrogels. **g** Azimuthal integral intensity distribution of PVA–CDs, PVA–CDs–S, and PVA–CDs–SP hydrogels. **h** Full width at half maximum (FWHM) and degree of orientation for PVA–CDs, PVA–CDs–S and PVA–CDs–SP hydrogels. **i** SAXS spectra of PVA–CDs, PVA–CDs–S, and PVA–CDs–SP hydrogels.

cycles, the energy dissipation slightly decreased (Fig. 5b), due to damage to some crystalline domains during cycling. The energy dissipation and the ratio of dissipated energy increased slightly with the number of cycles, which was attributed to the intrinsic cross-linking networks efficiently facilitating energy dissipation and preventing crack propagation. Furthermore, the consistent peak stress observed after 200 cycles indicated that hydrogel exhibited excellent stability and reproducibility (Fig. 5c). Subsequently, the swelling behavior of PVA–CDs–SP hydrogels was further investigated (Supplementary Fig. 25). The hydrogels reached swelling equilibrium after one week of immersion in water. Importantly, the ratio of swelling for PVA–CDs–SP hydrogel was −3%, respectively, indicating exhibit excellent swelling resistance, this property was attributed to the cross-linking of the nanocrystalline domains and reduced osmotic pressure, it was crucial for long-term stable underwater motion detection.

The conductivities of PVA, PVA–CA, PVA–CDs, PVA–CDs–S, and PVA–CDs–SP were measured using electrochemical impedance spectroscopy (Fig. 5d and Supplementary Fig. 26). The PVA–CDs hydrogel exhibited stable conductivity due to the planar arrangement of numerous $sp^2$ hybridized carbon atoms in CDs, facilitating the formation of π-conjugation within the carbon atom system. This conjugation system enabled electrons to move more freely between the CDs, thereby enhancing the conductivity of hydrogel. Additionally, π–π interactions between the CDs and the hopping migration of electrons further contribute to the overall conductivity. The subsequent introduction of ions and the formation of the ordered structure further enhanced the conductivity of hydrogel. This ordered structure facilitating the formation of more organized ion channels, allowing ions to move more efficiently. However, after salt dialysis, the conductivity of the hydrogel decreased. This decline was attributed to osmotic

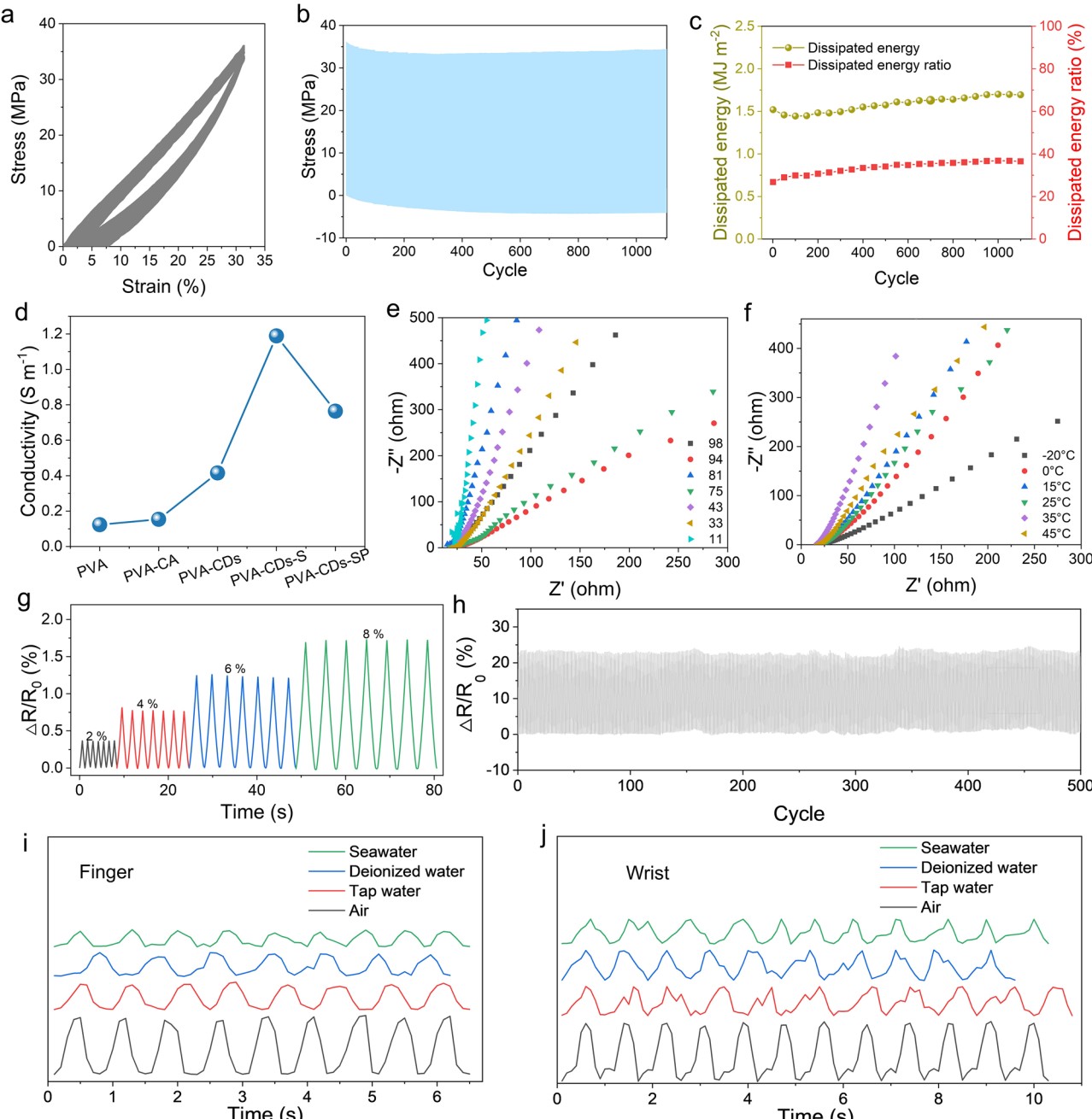

**Fig. 5 | Application for underwater sensing. a** and **b** Cyclic tensile test of the PVA–CDs–SP hydrogel under tension in the range of 0–30% over 1100 cycles. **c** The corresponding dissipated energy and dissipated energy ratio. **d** Conductivity of VA、PVA–CA, PVA–CDs, PVA–CDs–S, and PVA–CDs–SP hydrogels. Nyquist diagrams of PVA, PVA–CA, PVA–CDs, PVA–CDs–S, and PVA–CDs–SP hydrogels at different humidity environment (**e**) and temperatures (**f**). **g** Relative resistance variation of the hydrogel during cycles at different strains of 2%, 4%, 6%, and 8%. **h** Real-time resistance variation of the flexible sensor after undergoing 500 cycles of stretching to 80% strain. Real-time resistance signals during **i** finger bending and **j** wrist bending in various environments were monitored using flexible sensors based on PVA–CDs–SP hydrogel.

pressure differences inside and outside the hydrogel, leading to water loss, this reduction in water content negatively affected ion migration and then further decreased the conductivity of hydrogel.

Notably, the conductivity of the PVA–CDs–SP hydrogel showed only minor variations with changes in temperature and humidity (Fig. 5e, f and Supplementary Figs. 27, 28). It was attributed to the excellent swelling resistance of PVA–CDs–SP hydrogel, which maintained it to microstructure within a certain range. This stability allowed the hydrogel to efficiently resisted water absorption and evaporation during environmental fluctuations, leading in minimal changes in conductivity. Next, the sensing performance of the hydrogel as a strain

sensor was assessed. The gauge factor (GF) was 0.214 at small strain and 0.567 at large strain (Supplementary Fig. 29). The sensing stability of the hydrogel under various deformations was evaluated (Fig. 5g), where, its performance remained relatively stable during continuous loading and unloading. The cyclic stability of the hydrogel sensor through rapid loading–unloading experiments at 10% strain was also confirmed (Fig. 5h). The hysteresis curves remained nearly identical after 500 cycles, showing no significant degradation. It demonstrated the excellent reproducibility and repeatability of the hydrogel sensor.

The strong swelling resistance and stability of PVA–CDs–SP hydrogel made it suitable for underwater sensing applications.

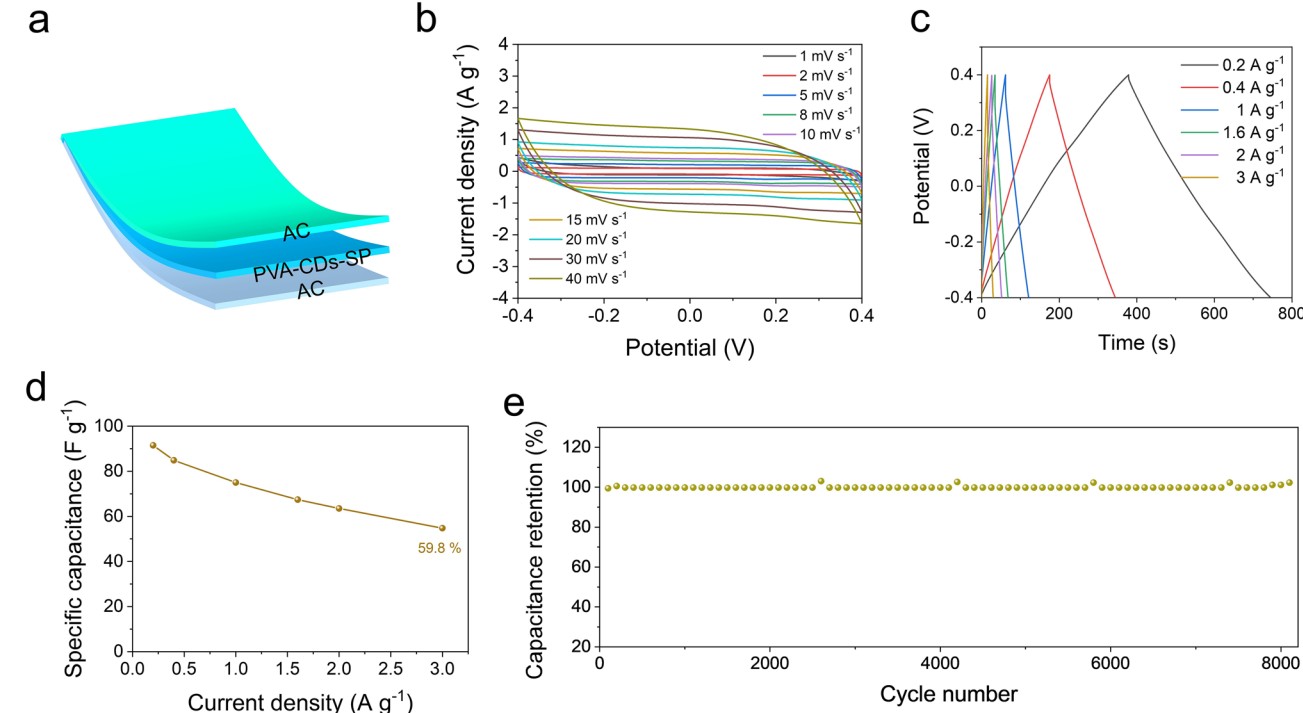

**Fig. 6 | Electrochemical performance of the supercapacitor using the PVA–CDs–SP hydrogel as a quasi-solid-state electrolyte. a** Schematic illustration of the assembled supercapacitor configuration. **b** C–V curves at different scan rates, showing ideal capacitive behavior. **c** GCD curves at various current densities. **d** Capacitance retention at different current densities, demonstrating rate capability. **e** Cycling stability of the device over 8000 charge–discharge cycles.

Subsequently, the sensing performances of the hydrogel strain sensors on fingers and wrists in various environments were tested (Fig. 5i, j). The sensors consistently provided stable signals in air, tap water, deionized water and seawater. This indicated that the sensor based on PVA–CDs–SP hydrogel was suitable for diverse environments, facilitating long-term monitoring of underwater motion.

### Electrochemical behavior in supercapacitors
To further explore the practical applications of the PVA–CDs–SP hydrogel, we assembled a quasi-solid-state supercapacitor by integrating the hydrogel as the electrolyte and separator, while using activated carbon (AC) as the electrode material (Fig. 6a). As shown in Fig. 6b, the cyclic voltammetry (C–V) curves at various scan rates exhibited nearly rectangular shapes with minimal distortion even at high scan rates, indicating excellent electrochemical reversibility and ideal capacitive behavior.

The galvanostatic charge–discharge (GCD) curves at different current densities (Fig. 6c) displayed highly symmetrical isosceles triangle shapes, characteristic of electric double-layer capacitors. The specific capacitance, calculated from the GCD profiles, reached 91.5 F g$^{-1}$ at a current density of 0.2 A g$^{-1}$. Even when the current density was increased by 15 times, the device retained 59.8% of its capacitance (Fig. 6d), demonstrating high rate capability. Furthermore, the supercapacitor based on the PVA–CDs–SP hydrogel exhibited excellent long-term cycling stability, maintaining nearly 100% capacitance retention after 8000 charge–discharge cycles (Fig. 6e). These results clearly demonstrated that the PVA–CDs–SP hydrogel not only offered mechanical robustness but also served as an efficient quasi-solid-state electrolyte for high-performance supercapacitors. In addition, the strong ionic conductivity, water-retention ability and structural stability of the hydrogel contributed to maintaining a stable ionic environment at the electrode–electrolyte interface, which was crucial for reliable energy storage performance under flexible or wearable conditions. Its adaptability to deformation and potential for integration into stretchable electronics further highlighted its potential in next-generation soft energy storage systems.

### Discussion
In this study, we report a strategy for the formation of crystallization domains induced by CDs and the preparation of strong hydrogels (PVA–CDs–SP) based on the "pinning effect". This approach led to the development of hydrogels with strong properties, including high strength (-156 MPa), excellent toughness (-225.2 MJ m$^{-3}$), and an ordered structure. Notably, the mechanical properties of the hydrogels could be easily tailored by adjusting the crystallization domains. Importantly, this strategy found to be applicable to various combinations of PVA and various CDs. Additionally, the hydrogels exhibited excellent conductivity and anti-swelling properties, making it suitable for underwater human motion monitoring and supercapacitor using the PVA–CDs–SP hydrogel as a quasi-solid-state electrolyte. Given the favorable dispersion and the adjustable $sp^2$ hybridized carbon structures of CDs, this strategy enables the construction of robust hydrogels through CDs induced crystallization domains and synergized orientation regulation based on "pinning effect" using for underwater sensing. This strategy enables the use of otherwise fragile hydrogels in a wider range of applications and facilitating their practical implementation.

### Methods
#### Materials
PVA (PVA-1799, 27.0–34.0 mPa s), citric acid (CA, 99.5%), sodium citrate (98%), malic acid (98%), and tannic acid (95%) were obtained from Shanghai Adamas Reagent Co., Ltd. Eucalyptus bark was obtained from a local market.

#### Preparation of citric acid carbon dots (CDs)
Citric acid CDs were prepared using a hydrothermal method. First, citric acid (1 g) was dissolved in 100 mL of deionized water and reacted

at 180 °C for 8 h. The solution was then allowed to cool to room temperature to obtain the CDs.

## Preparation of PVA hydrogel

PVA was dissolved in deionized water and stirred at 90 °C for 3 h to obtain PVA solutions with different concentrations. The homogeneous solution was then poured into silica molds, frozen at −20 °C for 12 h, thawed at room temperature for 2 h, and subjected to three freeze–thaw cycles to obtain the PVA hydrogel.

## Preparation of PVA–CA$_{10\%}$ hydrogel

PVA was dissolved in a 10% citric acid (CA) solution and stirred at 90 °C for 3 h to obtain a PVA–CA$_{10\%}$ solution. The resulting mixture was poured into a silicone mold, frozen at −20 °C for 12 h, and then thawed at room temperature for 2 h. This freeze–thaw cycle was repeated three times to obtain the PVA–CA$_{10\%}$ hydrogel. Unless otherwise specified in the main text, PVA–CA refers to the PVA$_{20\%}$–CA$_{10\%}$ composition.

## Preparation of PVA$_n$-CDs$_m$ hydrogel

PVA was dissolved in the CDs solution and stirred at 90 °C for 3 h to obtain PVA$_n$–CDs$_m$ solutions with different concentrations. The homogeneous solution was then poured into silica molds, frozen at −20 °C for 12 h, thawed at room temperature for 2 h, and subjected to three freeze–thaw cycles to obtain the PVA$_n$–CDs$_m$ hydrogel. Here, $n$ represents the concentration of PVA in the PVA–CDs hydrogel, $m$ represents the concentration of the CDs solution.

Unless otherwise specified in the main text, PVA–CDs refers to the PVA$_{20\%}$–CDs$_{10\%}$ composition.

## Preparation of PVA–CDs–S hydrogel

The PVA–CDs hydrogels were soaked in a 2 mol L$^{-1}$ sodium citrate solution for 12 h to obtain PVA–CDs–S hydrogel.

## Preparation of PVA–CDs–SP$_x$ hydrogel

PVA–CDs–S hydrogel was then gradually stretched to 100–600% of its original length and subsequently soaked in a sodium citrate solution for another 12 h to obtain PVA–CDs–SP$_x$ hydrogel. Here, $X$ represents the PVA–CDs–S hydrogel stretched to different strain levels. The hydrogel codes and the weight fractions of each component in the hydrogels were summarized in Supplementary Table 1. Different CDs prepared with malic acid, tannic acid and eucalyptus bark to verify the universality. All parameters were consistent with those used in the preparation of PVA–CDs–SP hydrogels.

## Preparation of supercapacitors

Mixing activated carbon (AC), acetylene black, and polyvinylidene fluoride (PVDF) in a mass ratio of 8:1:1, then, dissolved the mixture with a small amount of isopropyl alcohol to form a uniform slurry, then apply the slurry to nickel foam. The activated carbon electrode was dried overnight at 80 °C and pressed into sheets under a pressure of 18 MPa. The symmetrical supercapacitors consist of two opposing carbon electrodes, which are sandwiched between a PVA–CDs–SP hydrogel (15 mm × 7.5 mm × 2 mm) that serves as both the electrolyte and separator.

## Mechanical performance test

The mechanical properties of the hydrogels were evaluated using an ETM-10B Universal Mechanical Testing Machine (Shenzhen Vance Testing Machine Co., Ltd.) equipped with a 500 N load cell, at a room temperature of 25 ± 3 °C and a relative humidity of 70 ± 5%. The dimensions of the PVA, PVA–CA, and PVA–CDS hydrogel samples are 10 mm in width, 2 mm in thickness, and 40 mm in length. The ends of the samples were wrapped in newspaper to minimize slippage, and the tensile speed was set to 100 mm min$^{-1}$ unless

otherwise specified. For cyclic tensile testing, the hydrogel samples were stretched to a fixed strain at a rate of 80 mm min$^{-1}$, followed by immediate unloading at the same rate, without waiting for the sample to equilibrate before the next cycle. Young's modulus was determined from the initial slope of the stress−strain curve, toughness was estimated from the area under the curve, and energy dissipation was calculated from the area between the loading and unloading curves.

## Electrochemical property test

The electrical and sensing properties of the hydrogels were evaluated using an electrochemical workstation (CHI660E, Shanghai, China). The impedance spectrum was recorded over a frequency range of 0.1–100,000 Hz, and the relative change in resistance was calculated as $(R_0−R)/R_0$, where $R$ and $R_0$ represent the resistance after applying strain and the initial resistance, respectively. Conductivity ($\sigma$) was calculated as

$$\sigma = \frac{I}{RS} \tag{1}$$

where $I$, $R$, and $S$ represent the current, resistance, and cross-sectional area of the hydrogel, respectively.

## Fourier-transform infrared (FT-IR)

Characterization was performed using a Thermo Scientific iN10 Fourier transform infrared spectrometer.

## Raman spectroscopy analysis

Raman spectra of the samples were acquired using a Thermo Fisher DXR 2xi confocal Raman spectrometer with a 532 nm laser at 5 mW laser intensity.

## X-ray diffraction (XRD) analysis

X-ray diffraction (XRD) analysis of the samples on the slides was performed using a Bruker D8 Advance XRD system with Cu Kα radiation ($\lambda = 1.5418$ Å) at a scanning speed of 10° min$^{-1}$.

## Scanning electronic microscopy (SEM) analysis

The hydrogel was freeze-dried, and then its morphology was observed using scanning electron microscopy (SEM, ZEISS GeminiSEM 300, Germany). The freeze-dried hydrogel was directly adhered to conductive adhesive and coated for 45 s at 10 mA using a Quorum SC7620 sputter coater.

## Polarized light microscope testing

The crystalline morphology of the sample was observed using a polarized light optical microscope (OLYMPUS GX71).

## Small-angle X-ray scattering (SAXS) measurements

SAXS measurements were conducted using an Anton Paar SAX-Spoint2.0 with a Cu target (Kα radiation, $\lambda = 0.154184$ nm) and an X-ray wavelength ($\lambda$) of 1.53 nm. The sample-to-detector distance was 300 mm, and the signal was collected using a 2D hybrid photon counting detector (EIGER R 1M). The scattered data were analyzed using Fit 2D software, and the SAXS 2D patterns were integrated over a fan-shaped region to obtain the one-dimensional scattering profile. The scattering vector ($q$) was calculated using the equation:

$$q = \frac{4\pi \sin \theta}{\lambda} \tag{2}$$

where $\lambda$ is the X-ray wavelength and $\theta$ is the scattering angle. The one-dimensional curve was integrated, and the average distance between

adjacent crystalline domains was then calculated using Bragg's law:

$$L = \frac{2\pi}{q_{max}} \qquad (3)$$

where $q_{max}$ is the value of $q$ at the maximum scattering intensity.

## X-ray photoelectron spectroscopy (XPS)
PVA, PVA–CA, PVA–CDs, PVA–CDs–S and PVA–CDs–SP were measured by X-ray photoelectron spectroscopy (XPS) using an X-ray photo-electron spectrometer (Thermo Fisher Scientific, USA).

## Low-field nuclear magnetic resonance (NMR) analysis
The proton spin–spin relaxation time ($T_2$) of the hydrogel was measured using a MesoMR23-060H-I nuclear magnetic resonance imaging analy-zer, with a proton resonance frequency of 21 MHz (0.5 T), a resonance frequency of 23 MHz, a coil diameter of 25 mm, and a magnet tem-perature of 32 °C. The correlation time ($\tau_c$) of water molecular motion was calculated using the Bloembergen–Purcell–Pound equation, thus enabling the quantitative analysis of water dynamics:

$$\frac{1}{T_2} = \frac{C}{2}\left(3\tau_c + \frac{5\tau_c}{1+\omega_0^2\tau_c^2} + \frac{2\tau_c}{1+4\omega_0^2\tau_c^2}\right) \qquad (4)$$

Here, $C$ is the water constant ($5.33 \times 10^9\,S^{-2}$), and $\omega_0$ is the Larmor frequency.

## Differential scanning calorimetry (DSC) analysis
The gelation point, melting peak of the crystallization domain, and enthalpy change of the hydrogels were determined using a NETZSCH DSC 3500. First, the mass ($M$) of the freeze-dried sample was measured, and the sample was then heated from −80 to 250 °C at a rate of 10 °C min$^{-1}$ in a nitrogen atmosphere. The peak observed between 200 and 250 °C was the melting peak of the hydrogel crystallization domain, and the area under this peak was denoted as $H_{crystalline}$. The crystallinity of the crystallization domain can be calculated as

$$m_{crystalline} = M\,H_{crystalline}/H^0_{crystalline} \qquad (5)$$

where, $H^0_{crystalline} = 138.6\,J\,g^{-1}$, represents the 100 wt% enthalpy of melting. The crystallinity ($X_{dry}$) of the dry sample can then be calcu-lated as

$$X_{dry} = m_{crystalline}/M \qquad (6)$$

## Transmission electron microscope (TEM) test
TEM images were acquired using a transmission electron microscope, the FEI Talos F200S (USA), at an accelerating voltage of 200 kV.

# Data availability
All data supporting the findings of this study are available within the article and the Supplementary Information file. All data are available on request from the corresponding authors.

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

## Acknowledgements

This work was financially supported by the National Natural Science Foundation of China (No. 32171884), the Applied Basic Research Program of Yunnan Province (No. 202301AS070041), and Major Science and Technology Project of Yunnan Province (202402AE090027). L.Y. acknowledges Candidates for the Young and Middle-Aged Academic Leaders of Yunnan Province (202105AC160048) and the Ten Thousand Talent Program for Young Topnotch Talents of Yunnan Province (YNWR-QNBJ-2020-136). G.D. acknowledges the Yunnan Provincial Academician Workstation (YSZJGZZ-2020052) and the 111 Project (D21027). J.W. acknowledged the supported by the Yunnan Province Joint Agricultural Basic Research Project (202501BD070001-103). J.W. acknowledged the supported by the Scientific Research Fund project of Education Department of Yunnan Province (2025J0626).

## Author contributions

J.W. conceived and supervised the project. J.W. and H.H. have contributed to the conceptualization of the paper, as well as the development of its methodology and research design. J.W. and H.H. finished the manuscript. H.S., H.Y., X.D., Y.G., Y.C., F.K., and H.L. performed the experiments and characterizations. J.S., G.D., and L.Y. reviewed the paper. X.D. and Y.C. were responsible for data preparation and management. Y.C. and Y.G. conducted the analysis and prepared the display items. All authors discussed the results and commented on the manuscript.

## Competing interests

The authors declare no competing Interests.
