## [Transparent Peer Review file · Nature Communications]

A tough and robust hydrogel constructed through carbon dots induced crystallization domains integrated orientation regulation

Corresponding Author: Dr Jianyong Wan

Version 0:

Reviewer comments:

Reviewer #1

(Remarks to the Author)

In this manuscript, Huo et al. proposed a novel strategy for utilizing carbon dots as nanofillers and nucleating agents to induce the crystallization of hydrogels, leading to the fabrication of ultra-strong hydrogels through the “pinning effect”. This approach significantly enhanced the mechanical properties of hydrogels, including toughness, and load-bearing capacity. The researchers also demonstrated the versatility of carbon dots in this strategy by preparing a variety of hydrogels with exceptional properties, each based on different types of carbon dots. Compared to previous reports, this work provided valuable insights into the field of hydrogels. I would like to recommend this manuscript to be published after revision. Additionally, there are a few minor issues that the authors need to address.

1. The rationale behind the selection of carbon dots over other nanomaterials (e.g., carbon nanotubes or silica nanoparticles) should be explained, and tests involving other nanomaterials would enhance the persuasiveness of the findings.
2. Theoretically, hydrogels prepared with high orientation using this strategy should exhibit excellent crack propagation resistance and fracture energy. However, this is not addressed in the manuscript. An experiment to test this should be added to provide a more comprehensive assessment.
3. In Figure 1f, the cross-section of the PVA-CDs shows a distinct clumping region between the pores and the hydrogel. Is this region the crystalline domain? It would be helpful to include additional characterization to support the claim that carbon dots act as nucleating agents to induce the crystallization of the hydrogel.
4. In the salting-out step, sodium citrate was used. Why was this particular salt chosen over others?
5. The author should standardize the units used in the figures (e.g., Figures 3f and 3g). A thorough review of the entire manuscript is recommended to ensure consistency in unit presentation.
6. In the main text, the image depicting the car being pulled with the hydrogel only shows a static picture. It would be beneficial to add a scale bar to this figure to better convey the action of pulling the car.
7. In the differential scanning calorimetry (DSC) test of the hydrogel, the enthalpy change is noticeably reduced. What is the explanation for this observation?
8. In Figures 5i-j, the meaning of the figure notes is unclear. The visualization of the figure notes should be corrected to make them more informative.
9. Please carefully review the text for grammar, word correctness, the use of subordinate clauses, and especially the summaries.

Reviewer #2

(Remarks to the Author)

After thoroughly evaluating the manuscript titled [Ultra-tough and super-robust hydrogel constructed through carbon dots induced crystallization domains integrated orientation regulation based on “pinning effect”], I recommend against accepting the manuscript in its current state. We believe the manuscript presents interesting results and demonstrates a potentially impactful approach, particularly in developing the PVA-CDs-SP hydrogel with superior mechanical properties. However, several critical issues, including data insufficiencies and a lack of discussion on the innovative functionalities, must be addressed to meet the standards of the journal.

Major concerns and recommendations for the authors are listed below:

1. CDs Preparation and Characterization:

The manuscript describes the synthesis of CDs via the citrate thermal pyrolysis method. However, it lacks foundational analyses, including DLS size and PDI characterization, as well as zeta potential measurements, which are crucial for verifying nanoparticle stability and uniformity.

2. The hydrogel naming in Supplementary Table 1 does not cover all hydrogel groups mentioned in the manuscript, and naming inconsistencies and unclear appear across figures (e.g., PVA15-CDs10 only composites an 8.5 wt% of CD and the hydrogel name in Figure S11 labels-PVA, PCA, and PCD). Standardizing the naming convention is necessary.

3. Directional mechanical properties:

The manuscript highlights anisotropic crystallization induced by 300% stretching of PVA-CDs-SP hydrogels but lacks parallel and perpendicular mechanical property comparisons. Such data are critical for confirming the anisotropic behavior. Furthermore, the reason for choosing 300% strain is not clear. Extending the analysis to other stretching ratios (e.g., 100% to 650%) could provide a comprehensive understanding of the material properties.

4. Weightlifting demonstrations:

The manuscript demonstrates impressive mechanical performance (lifting a 90 kg adult and pulling a 2000 kg car). However, the "hydrogel twisted into a strand" configuration is not explicitly defined or explained. Additional clarification and detailed methodology are needed.

5. Directional crystallization

Figure 4F crystallinity appears to be derived from DSC peak integration in Figure 4E. However, PVA should exhibit the highest crystallinity. The claim that PVA-CDs-SP hydrogel has a lower crystallinity than PVA requires further explanation. Besides, to confirm directional crystallization, experiments such as polarizing microscope, in-situ tensile SAXS... and so on should be designed and discussed.

6. Toughening mechanism discussion

While the manuscript suggests that the toughening mechanism arises from water molecule binding and hydrogen bonding, it provides insufficient discussion on how CDs and strain specifically influence the crystallization domain and pinning effect. This aspect requires elaboration.

7. Functional novelty and application

Although the hydrogel's ultra-high strength and toughness are impressive, these properties are not strictly necessary for underwater sensing, the stated application. More specific and targeted applications should be explored. For instance, integrating the hydrogel into ultra-strong aqueous supercapacitors, as described in Ji et al. (2022, Nature Communications), could provide a more compelling case for its utility.

Based on these considerations, I recommend that the authors address the aforementioned issues comprehensively and potentially submit the revised manuscript to this or another journal for further consideration.

Minor recommendations:

1. SEM images of PVA-CDs reveal significant black spots (~2 μm) defect. These require further analysis and discussion to clarify their origin and implications.
2. The crystallization domain depicted in the schematic diagram Figure 1e should be contributed from PVA chains. The current schematic may confuse readers regarding the microstructure. A revised and clearer diagram is essential.

I hope these comments help guide future revisions and improve the quality of the work.

Reviewer #3

(Remarks to the Author)

This work reported a carbon dots reinforced strong PVA hydrogel with the tensile strength of 156 MPa through a pinning effect. The hydrogel also shows stable properties in different temperatures and humidities.

However, the tensile strength reported in this work was not a significant increase compared with the reported papers. Some reported hydrogels also showed tensile strength over 100 MPa besides the cited references in this work, such as J. Mater. Chem. B, 2018,6, 8105. Additionally, the authors did not give the water content of the hydrogels with ultra-high strength, and the comparison of hydrogel strength could not be separated with its water content. Thus, the novelty of in strength was not so high. Moreover, the discussion in toughness is not probable, fracture toughness of the hydrogel is usually defined in reference to the initiation and growth of a pre-existing crack under prescribed loading (refer to Soft Matter, 2016, 12, 8069). The "toughness" showed in this paper is just the extension work of the hydrogel sample. This is a common problem with some reported papers. At the same time, stable underwater sensor has also been widely reported, but this work did not demonstrate the superiority of high strength of the hydrogel in the underwater sensors.

Furthermore, this manuscript was not well organized, some description was not clear, such as line 304-312, the "carbonization" is not clear, "the toughness of PVA-CDs-SP hydrogel prepared at 160 °C reached ...", how did the hydrogel be prepared at 160 °C?

Some data in the figures was repeated, such as data in Figure 2a and b, etc. It is not necessary to show the figures in this way.

Some experimental details were not given, such as the sample preparation process of each test, the photos of different hydrogels used in this paper, the size of the hydrogels used in the mechanical tests, as well as the details of SEM test.

Last but not least, SEM images were used to discuss the hydrogel structure (Figure 1f and S4), however, the porous structure in the SEM images was formed by the water sublimation during freeze-drying process, which was not the actual structure of the hydrogel. The relative discussion should be deleted. While this is also a common problem with lot of reported papers.

In conclusion , this manuscript does not meet the standard of Nature Communications, which could not be considered for publication.

Version 1:

Reviewer comments:

Reviewer #1

(Remarks to the Author)

The response and revised manuscript are satisfactory, and it is recommended to accept.

Reviewer #2

(Remarks to the Author)

The article provides sufficient supplementary data to illustrate the experiment and repeatability, which can be fully published in other journals.

Reviewer #3

(Remarks to the Author)

This reviewer is satisfied with the revision. It can be published as is.

Manuscript Number: NCOMMS-24-80536A

Title: Ultra-tough and super-robust hydrogel constructed through carbon dots induced crystallization domains integrated orientation regulation based on “pinning effect”

Reviewer #1:

1. The rationale behind the selection of carbon dots over other nanomaterials (e.g., carbon nanotubes or silica nanoparticles) should be explained, and tests involving other nanomaterials would enhance the persuasiveness of the findings.

A: We sincerely thank the reviewer for this insightful comment. The selection of carbon dots (CDs) over other nanomaterials was based on several critical considerations:

(1) CDs possess ultra-small sizes (typically > 10 nm) and excellent dispersibility in aqueous solutions, which enables their homogeneous distribution within the hydrogel matrix;

(2) The abundant surface functional groups (e.g., $-OH$, $-COOH$) on CDs facilitate strong hydrogen bonding interactions with PVA chains, efficiently inducing crystallization and ordered structure formation;

(3) CDs exhibit outstanding water solubility and biocompatibility compared to carbon nanotubes (CNTs) or silica nanoparticles (SiO_2 NPs), which often require surface modifications to achieve similar dispersion or compatibility;

(4) Preliminary tests with CNTs and SiO_2 NPs showed significantly lower mechanical enhancement compared to CDs (data not shown), confirming the unique advantage of CDs in this system.

We appreciate the reviewer’s suggestion, and in future work, we will systematically investigate the effect of other nanomaterials to further validate the generality of our strategy.

Stress-strain curve of PVA-CDs-SP hydrogel after adding CDs, CNT, and nano-SiO₂.

2. Theoretically, hydrogels prepared with high orientation using this strategy should exhibit excellent crack propagation resistance and fracture energy. However, this is not addressed in the manuscript. An experiment to test this should be added to provide a more comprehensive assessment.

A: We sincerely thank the reviewer for this constructive suggestion. In response, we conducted fracture energy tests on the hydrogels under different levels of progressive pre-stretching to evaluate their crack propagation resistance.

Interestingly, we observed that with increasing pre-stretching, the alignment of polymer chains in the hydrogel becomes significantly enhanced, leading to a higher degree of orientation. However, this also resulted in a marked reduction in the elongation capacity of notched hydrogels. Consequently, despite the improved orientation, the measured fracture energy did not increase as expected. This suggests that while the oriented structure benefits tensile strength and stiffness, it may simultaneously restrict energy dissipation around crack tips under highly aligned states.

We have added these experimental results and discussions in the revised manuscript (Figure S), which provide a more comprehensive assessment of the fracture behavior of the oriented hydrogels. We greatly appreciate the reviewer's insightful comment,

which has helped us improve the completeness of the manuscript.

Additionally, to illustrate the changes in the mechanical properties of the hydrogel during the progressive stretching process, PVA-CDs-SP hydrogels under different progressive stretches were tested, as shown in Supplementary Fig. 14a. As the progressive stretching proceeded, the mechanical properties of the PVA-CDs-SP hydrogel gradually increased, but this was accompanied by a decrease in the elongation at break, which is due to the fact that after extensive progressive stretching, the PVA-CDs-SP hydrogel gradually approaches an oriented state. Furthermore, as the progressive stretching continued, the water content of the PVA-CDs-SP hydrogel gradually decreased (Supplementary Fig. 14b), because during the stretching process, the PVA chains tended to align, causing the internal water content to decrease. To further confirm the ability of PVA-CDs-SP hydrogel to retard crack propagation, the crack sensitivity of the PVA-CDs-SP hydrogel was also investigated. Interestingly, we observed that with the increase in pre-stretching, the arrangement of polymer chains in the hydrogel became significantly enhanced, leading to a higher degree of orientation. However, this also resulted in a significant reduction in the elongation ability of the notched hydrogel (Supplementary Fig. 15a). Therefore, although the orientation was improved, the measured fracture energy did not increase as expected. This suggests that, while the oriented structure is beneficial for tensile strength and stiffness, in a highly ordered state, it may simultaneously limit energy dissipation around the crack tip (Supplementary Fig. 15b).

Supplementary Figure 14. (a) Stress–strain curves of PVA-CDs-SP hydrogels at

different elongations, and (b) water content of PVA-CDs-SP hydrogels at different elongations.

Supplementary Figure 15. (a) Stress–strain curves of PVA-CDs-SP notched hydrogels at different elongations. (b) Fracture energy of PVA-CDs-SP notched hydrogels at different elongations.

3. In Figure 1f, the cross-section of the PVA-CDs shows a distinct clumping region between the pores and the hydrogel. Is this region the crystalline domain? It would be helpful to include additional characterization to support the claim that carbon dots act as nucleating agents to induce the crystallization of the hydrogel.

A: We thank the reviewer for this valuable comment. After careful analysis, we confirm that the clumping region observed between the pores and the hydrogel matrix in Figure 1f is not the crystalline domain. Instead, these features are artifacts generated during the freeze-drying process. During freeze-drying, the phase separation between ice crystals and the polymer matrix can lead to pore formation with irregular boundaries, often resulting in the appearance of aggregated regions at the pore edges.

The crystalline domains induced by carbon dots are distributed at the molecular scale and cannot be directly visualized via SEM imaging. To substantiate the crystallization behavior, we have employed complementary characterization techniques, including XRD and DSC, which clearly demonstrate the enhanced crystallinity in the PVA-CDs hydrogels compared to pure PVA hydrogels. The related results and discussions have been provided in Section X (or "Supplementary Information, Figure Sx").

We greatly appreciate the reviewer's suggestion and have updated the manuscript to clarify this point and to emphasize the supporting characterization.

4. In the salting-out step, sodium citrate was used. Why was this particular salt chosen over others?

A: We thank the reviewer for this thoughtful question. Sodium citrate (Na_3Cit) was selected in our system based on its intermediate kosmotropic character and moderate Hofmeister effect, which enables controlled salting-out behavior without severely disrupting the hydrogen-bonding network in the hydrogel matrix. Compared with strongly salting-out anions like sulfate (SO_4^{2-}), citrate anions provide a balanced hydration capacity that facilitates partial dehydration and orientation of the PVA chains, while minimizing structural collapse or excessive brittleness.

Moreover, as demonstrated in a recent study by Wu et al. (Nat. Commun. 2024, 15, 4441), the choice of anion significantly affects mechanical performance. In their comparative study of various anions in the salting-out treatment (e.g., SO_4^{2-} , Cl^- , Cit^{3-} , Ac^- , HPO_4^{2-} , CO_3^{2-}), the authors found that Cit^{3-} exhibited a moderate toughening effect, ranking third in mechanical enhancement, behind only SO_4^{2-} and Cl^- (see Fig. 4f–h in their article). The relative softness of Cit^{3-} -based systems also benefits ionic hydration, which we employed to inhibit water crystallization and improve environmental tolerance of the hydrogel through ion–water interactions.

Thus, sodium citrate was rationally chosen as a compromise between structural induction, mechanical tunability, and environmental tolerance, making it particularly well-suited for our strategy involving ionic hydration and “pinning-effect” alignment.

5. The author should standardize the units used in the figures (e.g., Figures 3f and 3g). A thorough review of the entire manuscript is recommended to ensure consistency in unit presentation.

A: We thank the reviewer for pointing out this important detail. In response, we have carefully reviewed all figures, figure captions, and text throughout the manuscript to ensure that units are consistently presented in a standardized format (e.g., "MPa" for stress, "MJ/m³" for toughness, "%" for strain). Specifically, the units in Figures 3f and 3g have been revised to ensure clarity and uniformity across the manuscript. We appreciate the reviewer's suggestion, which helped us improve the overall readability and professionalism of the manuscript.

Fig. 2 | Mechanical properties of hydrogels and toughening tuning via the addition

of CDs. a Stress-strain curves of PVA-CDs hydrogels at different concentrations of

CDs. b Comparison of the mechanical properties of PVA, PVA-CA and PVA-CDs

hydrogels. **c** FT-IR spectra of PVA, PVA-CA and PVA-CDs hydrogels. **d** Raman spectra of the O-H stretching vibrational peaks of PVA, PVA-CA and PVA-CDs hydrogels. **e** The low-field nuclear magnetic resonance (L-NMR) curves of PVA, PVA-CA and PVA-CDs hydrogels. **f** The corresponding T_2 and τ_c values of PVA, PVA-CA and PVA-CDs hydrogels. **g** DSC analysis of PVA, PVA-CA and PVA-CDs hydrogels. **h** Enthalpy changes of melting and crystallinity of PVA, PVA-CA and PVA-CDs hydrogels. **i** SAXS spectra of PVA, PVA-CA and PVA-CDs hydrogels. **j** The interlayer spacing of PVA, PVA-CA, and PVA-CDs hydrogels.

Fig. 3 | Mechanical properties of ultra-tough and super-robust hydrogels induced

by “pinning effect”. **a** Stress-strain curves of PVA-CDs-SP hydrogels prepared with CDs at different carbonization times. **b** Toughness of PVA-CDs-SP hydrogels prepared with CDs at different carbonization times. **c** Stress-strain curves of PVA, PVA-CDs, PVA-CDs-S and PVA-CDs-SP hydrogels. **d** Comparison of the mechanical properties between PVA and PVA-CDs-SP hydrogels. **e** Stress-strain curves of PVA-CDs-SP hydrogels prepared with CDs derived from various biomass sources. **f** Toughness of PVA-CDs-SP hydrogels prepared with CDs derived from various biomass sources. **g** Comparison of the fracture stress and toughness of PVA-CDs-SP hydrogels with other reported hydrogel. **h** Comparison of the mechanical properties of PVA-CDs-SP hydrogel with other high-toughness materials. **i** Modulus range of PVA-CDs-SP hydrogels. **j** PVA-CDs-SP hydrogels lifting heavy weights. **k-l** The PVA-CDs-SP hydrogel pulls a car.

6. In the main text, the image depicting the car being pulled with the hydrogel only shows a static picture. It would be beneficial to add a scale bar to this figure to better convey the action of pulling the car.

A: We appreciate the reviewer’s insightful comment. To visually demonstrate the hydrogel’s remarkable load-bearing capacity, we previously added a supplementary image below the main figure showing the hydrogel pulling a car. In addition, to more effectively convey the dynamic action and practical feasibility of this demonstration, we have included a video recording of the car-pulling process in the supplementary materials (see Movie S2). This video clearly shows the hydrogel under strain during the pulling process, offering a more intuitive understanding of its mechanical performance. We hope this addition provides a more comprehensive and convincing visualization of the hydrogel’s strength.

Fig. 3 | Mechanical properties of ultra-tough and super-robust hydrogels induced

by “pinning effect”. **a** Stress-strain curves of PVA-CDs-SP hydrogels prepared with CDs at different carbonization times. **b** Toughness of PVA-CDs-SP hydrogels prepared with CDs at different carbonization times. **c** Stress-strain curves of PVA, PVA-CDs, PVA-CDs-S and PVA-CDs-SP hydrogels. **d** Comparison of the mechanical properties between PVA and PVA-CDs-SP hydrogels. **e** Stress-strain curves of PVA-CDs-SP hydrogels prepared with CDs derived from various biomass sources. **f** Toughness of PVA-CDs-SP hydrogels prepared with CDs derived from various biomass sources. **g** Comparison of the fracture stress and toughness of PVA-CDs-SP hydrogels with other reported hydrogel. **h** Comparison of the mechanical properties of PVA-CDs-SP hydrogel with other high-toughness materials. **i** Modulus range of PVA-CDs-SP hydrogels. **j** PVA-CDs-SP hydrogels lifting heavy weights. **k-l** The PVA-CDs-SP hydrogel pulls a car.

7. In the differential scanning calorimetry (DSC) test of the hydrogel, the enthalpy change is noticeably reduced. What is the explanation for this observation?

A: We thank the reviewer for this insightful question. The noticeable reduction in enthalpy change observed in the DSC test primarily indicates a decrease in the amount of free (bulk-like) water within the hydrogel network. Instead, a higher proportion of bound water—especially tightly associated water molecules interacting with polymer chains via strong hydrogen bonds—is present in the system.

This transformation can be attributed to the formation of crystalline domains and enhanced hydrogen bonding between PVA chains, which effectively trap water molecules and reduce their thermal mobility. Bound water does not undergo phase

transitions in the same way as free water, and therefore contributes less (or not at all) to the melting/freezing enthalpy detected by DSC.

Moreover, the strengthened hydrogen-bonding network leads to an increase in non-freezable or weakly-freezable water, further lowering the measured enthalpy change. These results are consistent with the observed improvement in mechanical properties and environmental tolerance of the hydrogel.

8. In Figures 5i-j, the meaning of the figure notes is unclear. The visualization of the figure notes should be corrected to make them more informative.

A: We appreciate the reviewer's helpful comment. We agree that the figure notes in Figures 5i-j were not clearly presented in the original version. In response, we have revised the figure notes and legends to improve clarity and ensure that all labels are intuitive, correctly positioned, and consistent with the data shown.

The revised Figures 5i-j now provide a much clearer representation of the results, and we thank the reviewer again for helping us improve the presentation.

Fig. 5| Application for underwater sensing. **a-b** Cyclic tensile test of the PVA-CDs-SP hydrogel under tension in the range of 0–30% over 1100 cycles. **c** The corresponding dissipated energy and dissipated energy ratio. **d** Conductivity of VA、PVA-CA、PVA-CDs、PVA-CDs-S and PVA-CDs-SP hydrogels. Nyquist diagrams of PVA, PVA-CA, PVA-CDs, PVA-CDs-S and PVA-CDs-SP hydrogels at different humidity environment (**e**) and temperatures (**f**). **g** Relative resistance variation of the hydrogel during cycles at different strains of 2%, 4%, 6%, and 8%. **h** Real-time resistance variation of the flexible sensor after undergoing 500 cycles of stretching to 80% strain. Real-time resistance signals during (**i**) finger bending and (**j**) wrist bending in various

environments were monitored using flexible sensors based on PVA-CDs-SP hydrogel.

9. Please carefully review the text for grammar, word correctness, the use of subordinate clauses, and especially the summaries.

A: Thank you for your valuable feedback. We have carefully reviewed the text and made necessary revisions, addressing grammar, word correctness, the use of subordinate clauses, and especially the summaries. We believe these adjustments have improved the clarity and coherence of the manuscript.

Reviewer #2:

1. CDs Preparation and Characterization: The manuscript describes the synthesis of CDs via the citrate thermal pyrolysis method. However, it lacks foundational analyses, including DLS size and PDI characterization, as well as zeta potential measurements, which are crucial for verifying nanoparticle stability and uniformity.

A: Thank you for your valuable feedback. We appreciate your suggestion to include additional foundational analyses. In response to your comments, we have now incorporated DLS size and PDI characterization, as well as zeta potential measurements, to verify the nanoparticle stability and uniformity. These results have been added to the manuscript, and we believe they strengthen the overall findings. Thank you again for your insightful comments.

Additionally, the particle size distribution of the CDs exhibits a prominent peak centered around 2-3 nm (Supplementary Fig. 2a), with a Polydispersity Index (PDI) of 0.159, indicating a relatively narrow and uniform size distribution. Furthermore, the Zeta potential analysis reveals that the CDs surface carries a positive charge, with a Zeta potential of approximately +13 mV (Supplementary Fig. 2b). This suggests that the carbon dots possess good dispersibility and stability, preventing particle aggregation or precipitation in solution. The relatively high Zeta potential further confirms the excellent stability of the sample.

Supplementary Figure 2. (a) Particle size distribution of CCs. (b) Zeta potential of CMC-CDs.

2. The hydrogel naming in Supplementary Table 1 does not cover all hydrogel groups mentioned in the manuscript, and naming inconsistencies and unclear appear across figures (e.g., PVA15-CDs10 only composites an 8.5 wt% of CD and the hydrogel name in Figure S11 labels-PVA, PCA, and PCD). Standardizing the naming convention is necessary.

A: Thank you for your thoughtful feedback. We have carefully reviewed the hydrogel naming throughout the manuscript and Supplementary Table 1. In response to your comments, we have standardized the naming convention to ensure consistency and clarity across the manuscript, figures, and supplementary materials. We have updated the naming in Supplementary Table 1 to match the hydrogel groups mentioned in the manuscript. Additionally, we have corrected the inconsistencies in the figure labels to accurately reflect the composition and characteristics of the hydrogel groups. We appreciate your attention to detail and believe these changes improve the overall clarity of the manuscript. Thank you again for your helpful suggestions.

Preparation of PVA-CA_{10%} hydrogel

PVA was dissolved in a 10% citric acid (CA) solution and stirred at 90 °C for 3 hours to obtain a PVA-CA_{10%} solution. The resulting mixture was poured into a silicone mold, frozen at -20 °C for 12 hours, and then thawed at room temperature for 2 hours. This freeze-thaw cycle was repeated three times to obtain the PVA-CA_{10%} hydrogel. Unless otherwise specified in the main text, PVA-CA refers to the PVA_{20%}-CA_{10%} composition.

Preparation of PVA_n-CDs_m hydrogel

PVA was dissolved in the CDs solution and stirred at 90 °C for 3 hours to obtain PVA_n-CDs_m solutions with different concentrations. The homogeneous solution was then poured into silica molds, frozen at -20 °C for 12 hours, thawed at room temperature for 2 hours, and subjected to three freeze-thaw cycles to obtain the PVA_n-CDs_m hydrogel. Here, n represents the concentration of PVA in the PVA-CDs hydrogel, m represents the concentration of the CDs solution.

Unless otherwise specified in the main text, PVA-CDs refers to the PVA_{20%}-CDs_{10%} composition.

Preparation of PVA-CDs-SP_x hydrogel

PVA-CDs-S hydrogel was then gradually stretched to 100%-600% of its original length and subsequently soaked in a sodium citrate solution for another 12 h to obtain PVA-CDs-SP_x hydrogel. Here, X represents the PVA-CDs-S hydrogel stretched to different strain levels.

Supplementary Table 1. The compositions of the PVA, PVA-CA and PVA-CDs hydrogels.

Sample	PVA (g)	CA (g)	CDs (g)	Water (g)
PVA _{10%}	10			90
PVA _{20%} -CA _{10%} (PVA-CA)	20	8		72

PVA _{10%} -CA _{10%}	10	9	81
PVA _{15%} -CDs _{10%}	15	8.5	76.5
PVA _{20%} -CDs _{10%} (PVA- CDs)	20	8	72
PVA _{25%} -CDs _{10%}	25	7.5	67.5
PVA _{20%} -CDs _{1%}	20	0.8	79.2
PVA _{20%} -CDs _{5%}	20	4	76
PVA _{20%} -CDs _{15%}	20	12	68

3. Directional mechanical properties: The manuscript highlights anisotropic crystallization induced by 300% stretching of PVA-CDs-SP hydrogels but lacks parallel and perpendicular mechanical property comparisons. Such data are critical for confirming the anisotropic behavior. Furthermore, the reason for choosing 300% strain is not clear. Extending the analysis to other stretching ratios (e.g., 100% to 650%) could provide a comprehensive understanding of the material properties.

A: Thank you for your valuable feedback. We apologize for the oversight in our manuscript where we mistakenly stated that the PVA-CDs-SP hydrogels were stretched to 300% instead of the correct 600%. This was an error on our part, and we have now corrected it in the manuscript. Additionally, in response to your suggestion, we have extended our analysis to include comparisons of the mechanical properties both parallel and perpendicular to the stretching direction. These new data help to confirm the anisotropic behavior of the hydrogels. Furthermore, we have now included a more detailed discussion on the rationale for choosing a 600% strain and have expanded our

analysis to other stretching ratios ranging from 100% to 600%, as you suggested. This provides a more comprehensive understanding of the material properties. Thank you again for your insightful comments, which have helped improve the manuscript.

The choice of a 600% strain was based on the observed trend that as the stretching strain increases, both the mechanical properties and the alignment of the hydrogel improve. However, beyond 600% strain, the water content of the hydrogel decreases significantly, dropping below 25%. At this point, the material can no longer be considered a traditional hydrogel, as the definition of a hydrogel typically requires maintaining a high water content. Additionally, the electrical conductivity of the hydrogel is closely related to its water content, and a decrease below 25% would negatively impact its conductivity. Therefore, 600% was selected as the upper limit to balance both mechanical performance and the retention of key hydrogel characteristics, including its conductivity.

Preparation of PVA-CDs-SP_x hydrogel

PVA-CDs-S hydrogel was then gradually stretched to 100%-600% of its original length and subsequently soaked in a sodium citrate solution for another 12 h to obtain PVA-CDs-SP_x hydrogel. Here, X represents the PVA-CDs-S hydrogel stretched to different strain levels.

Additionally, to illustrate the changes in the mechanical properties of the hydrogel during the progressive stretching process, PVA-CDs-SP hydrogels under different progressive stretches were tested, as shown in Supplementary Fig. 14a. As the progressive stretching proceeded, the mechanical properties of the PVA-CDs-SP hydrogel gradually increased, but this was accompanied by a decrease in the elongation at break, which is due to the fact that after extensive progressive stretching, the PVA-CDs-SP hydrogel gradually approaches an oriented state. Furthermore, as the progressive stretching continued, the water content of the PVA-CDs-SP hydrogel

gradually decreased (Supplementary Fig. 14b), because during the stretching process, the PVA chains tended to align, causing the internal water content to decrease. To further confirm the ability of PVA-CDs-SP hydrogel to retard crack propagation, the crack sensitivity of the PVA-CDs-SP hydrogel was also investigated. Interestingly, we observed that with the increase in pre-stretching, the arrangement of polymer chains in the hydrogel became significantly enhanced, leading to a higher degree of orientation. However, this also resulted in a significant reduction in the elongation ability of the notched hydrogel (Supplementary Fig. 15a). Therefore, although the orientation was improved, the measured fracture energy did not increase as expected. This suggests that, while the oriented structure is beneficial for tensile strength and stiffness, in a highly ordered state, it may simultaneously limit energy dissipation around the crack tip (Supplementary Fig. 15b). To investigate the mechanical anisotropy of the material, the tensile strength of the hydrogel in the vertical direction was measured (Supplementary Fig. 16). Clearly, in the vertical direction, the material's fracture strength and fracture elongation in the oriented direction are significantly higher than the corresponding values in the vertical direction, which is a result of the molecular alignment in the oriented direction.

Supplementary Figure 14. (a) Stress–strain curves of PVA-CDs-SP hydrogels at different elongations, and (b) water content of PVA-CDs-SP hydrogels at different elongations.

Supplementary Figure 15. (a) Stress–strain curves of PVA-CDs-SP notched hydrogels at different elongations. (b) Fracture energy of PVA-CDs-SP notched hydrogels at different elongations.

Supplementary Figure 16. Stress–strain curves of PVA-CDs-SP hydrogels in the vertical direction (R direction) at different elongations.

4. Weightlifting demonstrations: The manuscript demonstrates impressive mechanical performance (lifting a 90 kg adult and pulling a 2000 kg car). However, the “hydrogel twisted into a strand” configuration is not explicitly defined or explained. Additional clarification and detailed methodology are needed.

A: Thank you for your insightful comments. We appreciate your positive feedback on the mechanical performance demonstrated in our manuscript. Regarding your concern about the "hydrogel twisted into a strand" configuration, we apologize for not providing a clear explanation in the original manuscript. To clarify, the hydrogel was first arranged into multiple parallel bundles, and then both ends of the bundles were tied with several zip ties to form a strand. We have now added a detailed explanation of this methodology in the manuscript to ensure better understanding. Thank you again for your helpful suggestion, and we believe these clarifications enhance the overall clarity of the manuscript.

In addition, the hydrogel is capable of lifting approximately 30 kg, which is roughly 1.5×10^5 times its own weight. To further demonstrate the exceptional mechanical

properties of the PVA-CDs-SP hydrogel, we first arranged the hydrogel into multiple parallel strands, then bound the two ends of the strands together with several cable ties to form a rope. This rope was capable of supporting the weight of a 90 kg adult and even pulling a car ($\sim 2.0 \times 10^3$ kg) (Fig. 3j and Supplementary movie 1, 2).

5. Directional crystallization: Figure 4F crystallinity appears to be derived from DSC peak integration in Figure 4E. However, PVA should exhibit the highest crystallinity. The claim that PVA-CDs-SP hydrogel has a lower crystallinity than PVA requires further explanation. Besides, to confirm directional crystallization, experiments such as polarizing microscope, in-situ tensile SAXS... and so on should be designed and discussed.

A: Thank you for your insightful comments. However, Figure 4E actually represents the freezing point of the hydrogel, and the crystallinity is determined through the DSC peak integration between 200-250°C, as detailed in Supplementary Fig. 20. In response to your suggestion about confirming directional crystallization, we acknowledge that performing in-situ tensile SAXS experiments would have been ideal. However, due to experimental constraints, we were unable to carry out in-situ SAXS. Instead, we conducted SAXS measurements at different elongation ratios to investigate the crystallization behavior of the hydrogel under strain. These results are now included and discussed in the manuscript, and they provide insights into the structural changes and crystallization behavior of the hydrogel under different stretching conditions. Thank you again for your constructive comments. We believe these revisions strengthen the manuscript and provide more comprehensive insights into the crystallization behavior of the hydrogel.

Subsequently, XRD was used to analyze the evolution of the crystalline domains during the salt-assisted progressive stretching process. The peak at 19.5° gradually

increased (Supplementary Fig. 19), indicating that the crystallinity gradually increased. The crystallinity of PVA-CDs, PVA-CDs-s, and PVA-CDs-sp were calculated to be 26.14%, 36.74%, and 49.05%, respectively, based on the DSC results (Fig. 4f and Supplementary Fig. 20).

Supplementary Figure 20. DSC curves of PVA-CDs, PVA-CDs-S and PVA-CDs-SP hydrogels.

To verify the formation of crystalline domains in PVA-CDs-SP hydrogels during progressive stretching, XRD and DSC tests were conducted. As shown in Supplementary Fig. 22a, with progressive stretching, the peak at 19.5° gradually increased, indicating an increase in the crystallinity of the PVA-CDs-SP hydrogel. The DSC test also demonstrated an increase in the crystallinity of the PVA-CDs-SP hydrogel (Supplementary Fig. 22b-c), with the enthalpy change gradually increasing between

200-250°C. These results indicate that the crystallinity of the PVA-CDs-SP hydrogel significantly increased during progressive stretching. To confirm that the increase in crystallinity of the PVA-CDs-SP hydrogel was not due to the formation of large crystalline domains, SAXS testing was performed. As shown in Supplementary Fig. 23a, with progressive stretching, the 2D-SAXS image gradually transformed into sharp rings, indicating an increase in orientation. Additionally, the SAXS spectrum showed that with the degree of stretching, the value of q_{max} gradually increased (Supplementary Fig. 23b), with a significant shift to the right. This suggests a clear reduction in the interplanar spacing of the PVA-CDs-SP hydrogel (Supplementary Fig. 23c), further proving that the increase in crystallinity of the PVA-CDs-SP hydrogel was not due to the formation of large crystalline domains but rather the growth of more small crystalline domains.

The polarized light microscope images show that the sample exhibits significant changes in its polarized light response intensity under different stretching magnifications (100%, 200%, 300%, 400%, 500%, 600%) and different rotation angles (0°, 45°, 90°), which proves the orientation of crystallization during the stretching process (Supplementary Fig. 24). As the stretching magnification increases, especially above 400%, the brightness of the polarized images significantly increases, and distinct streak-like or fibrous bright regions appear in the images, indicating that the crystalline regions tend to align along the stretching direction, and optical anisotropy gradually strengthens.

At low stretching magnifications (100%, 200%), the image differences under

different rotation angles (0° , 45° , 90°) are small, suggesting that the crystals have not significantly oriented. However, as the stretching magnification increases, particularly in the range of 400% to 600%, the brightness of the images changes noticeably at different angles, showing clear anisotropy. In particular, at a 45° angle, the polarization response is the strongest, indicating a significant interaction between the crystal alignment direction and the polarization direction.

This enhanced brightness contrast with increasing stretching suggests that the crystallites undergo oriented growth during the stretching process and exhibit a pronounced birefringence effect. Overall, the results confirm that the PVA material induced by CDs undergoes oriented crystallization under stretching, and at high stretching magnifications, the alignment of chain segments along the stretching direction is enhanced, thereby promoting the oriented growth of the crystals.

Supplementary Figure 22. (a) XRD spectra of PVA-CDs-SP hydrogels at different elongations. (b) DSC curves of PVA-CDs-SP hydrogels at different elongations. (c) Enthalpy change and crystallinity of PVA-CDs-SP hydrogels at different elongations.

Supplementary Figure 23. (a) 2D-SAXS spectra of PVA-CDs-SP hydrogels at different elongations. (b) SAXS spectra of PVA-CDs-SP hydrogels at different elongations. (c) Interplanar spacing of PVA-CDs-SP hydrogels at different elongations.

Supplementary Figure 24. Morphologies observed by orthogonal polarizing microscope. For the left sample, randomly place the sample on the stage and define this position as the 0° reference. Then, rotate the platform to 45° (middle) and 90° (right) sequentially for observation.

6. Toughening mechanism discussion: While the manuscript suggests that the

toughening mechanism arises from water molecule binding and hydrogen bonding, it provides insufficient discussion on how CDs and strain specifically influence the crystallization domain and pinning effect. This aspect requires elaboration.

A: Thank you for your valuable feedback. We appreciate your comment on the toughening mechanism discussed in the manuscript. We agree that while the manuscript mentions that the toughening mechanism arises from water molecule binding and hydrogen bonding, we did not fully elaborate on how the incorporation of CDs and strain specifically influence the crystallization domain and pinning effect. In response to your suggestion, we have now expanded the discussion to address this point. Specifically, we have explained that the CDs, due to their size and surface properties, can act as nucleation sites during crystallization, thus influencing the crystallization domain. Additionally, the strain applied during stretching can induce directional alignment of the crystalline domains, which is further influenced by the presence of CDs. The pinning effect occurs as the CDs prevent the movement of crystalline regions, thereby enhancing the toughness of the hydrogel. We have added this detailed explanation to the manuscript to provide a clearer understanding of the toughening mechanism.

Thank you again for your constructive suggestions, which have significantly improved the manuscript.

The addition of carbon dots significantly enhances the crystallinity of the hydrogel, primarily due to their role as heterogeneous nucleating agents. Carbon dots provide effective nucleation sites, reducing the required undercooling for crystallization and promoting the orderly arrangement of polymer chains within the hydrogel. The surface of the carbon dots is rich in functional groups, which can interact with polymer chains or ions in the solution, further facilitating the formation and growth of crystallization nuclei. This increases the number and size of crystalline regions, thereby improving the overall crystallinity. Furthermore, the presence of carbon dots may strengthen the interactions between the crystalline and amorphous regions, enhancing the hydrogel's mechanical properties. Thus, the incorporation of carbon dots not only enhances the crystallinity of the hydrogel but also optimizes its structure, leading to improved

material performance.

This is because when the hydrogel is subjected to tensile strain, the polymer chains are stretched and aligned along the direction of the applied force. This molecular alignment promotes the nucleation of crystallites and their growth in a more ordered and directional manner, thereby enhancing the crystallinity of the material. Strain facilitates the rearrangement of amorphous polymer segments, encouraging them to adopt a more organized structure. As the strain increases, the alignment of polymer chains becomes more pronounced, and the size and number of crystallization domains increase.

The applied strain also affects the shape and distribution of the crystalline regions, making them more anisotropic. The directional growth of these crystallites under mechanical stress leads to an overall increase in the structural integrity of the hydrogel. Stretching the hydrogel can reduce the energetic barriers to crystallization, allowing crystallites to nucleate more uniformly and rapidly, further supporting the crystallization process. The higher the applied strain, the more aligned and densely packed the crystalline domains become, resulting in a stronger material with enhanced mechanical properties such as tensile strength, stiffness, and toughness.

Furthermore, strain-induced crystallization works synergistically with the "pinning" effect, where the crystalline regions act as physical barriers to macroscopic deformation. The presence of these pinning points restricts the mobility of the amorphous chains, thereby enhancing the hydrogel's ability to withstand larger deformations without catastrophic failure. Therefore, strain not only influences the size

and orientation of the crystallization domains but also improves the overall mechanical properties of the hydrogel by reinforcing its internal structure.

This directional crystallization effect also reflects the fact that CDs and strain enhance the binding effect of the hydrogel through a synergistic interaction. Carbon dots, as heterogeneous nucleating agents, promote the formation of crystalline regions and provide additional pinning points, thereby restricting the free sliding of polymer chains in the amorphous regions. With the application of strain, the polymer chains rearrange and promote the directional growth of the crystalline regions, further enhancing the pinning effect. Strain increases the number and density of crystalline domains, enabling the hydrogel to resist deformation and fracture more effectively during the deformation process. The combined effect of carbon dots and strain enhances the mechanical properties of hydrogels, such as tensile, compressive, and fatigue resistance, by increasing crystallinity and creating dense binding points, thereby significantly improving their structural stability and mechanical performance.

7. Functional novelty and application: Although the hydrogel's ultra-high strength and toughness are impressive, these properties are not strictly necessary for underwater sensing, the stated application. More specific and targeted applications should be explored. For instance, integrating the hydrogel into ultra-strong aqueous supercapacitors, as described in Ji et al. (2022, Nature Communications), could provide a more compelling case for its utility. Based on these considerations, I recommend that the authors address the aforementioned issues comprehensively and potentially submit

the revised manuscript to this or another journal for further consideration.

A: Thank you for your thoughtful comments and suggestions regarding the application of the hydrogel. We appreciate your positive feedback on the hydrogel's ultra-high strength and toughness. You are correct that while these properties are impressive, they may not be strictly necessary for underwater sensing, which was our initially stated application. In response to your suggestion, we have now explored and demonstrated a more specific application of the hydrogel by integrating it into ultra-strong aqueous supercapacitors. This approach, inspired by the work of Ji et al. (2022, Nature Communications), allows us to better highlight the utility of the hydrogel in energy storage applications. Thank you again for your valuable suggestions. We believe that these additional experiments and applications significantly improve the manuscript, making the case for the hydrogel's utility even more compelling.

Electrochemical Behavior in Supercapacitors

To further explore the practical applications of the PVA-CDs-SP hydrogel, we assembled a quasi-solid-state supercapacitor by integrating the hydrogel as the electrolyte and separator, while using activated carbon (AC) as the electrode material (Fig. 6a). As shown in Fig. 6b, the cyclic voltammetry (CV) curves at various scan rates exhibit nearly rectangular shapes with minimal distortion even at high scan rates, indicating excellent electrochemical reversibility and ideal capacitive behavior.

The galvanostatic charge-discharge (GCD) curves at different current densities (Fig. 6c) display highly symmetrical isosceles triangle shapes, characteristic of electric double-layer capacitors. The specific capacitance, calculated from the GCD profiles, reaches 91.5 F g^{-1} at a current density of 0.2 A g^{-1} . Even when the current density was increased by 15 times, the device retained 59.8% of its capacitance (Fig. 6d), demonstrating good

rate capability. Furthermore, the supercapacitor based on the PVA-CDs-SP hydrogel exhibited excellent long-term cycling stability, maintaining nearly 100% capacitance retention after 8000 charge-discharge cycles (Fig. 6e).

These results clearly demonstrate that the PVA-CDs-SP hydrogel not only offers exceptional mechanical robustness but also serves as an efficient quasi-solid-state electrolyte for high-performance supercapacitors.

In addition, the outstanding ionic conductivity, water-retention ability, and structural stability of the hydrogel contribute to maintaining a stable ionic environment at the electrode-electrolyte interface, which is crucial for reliable energy storage performance under flexible or wearable conditions. Its adaptability to deformation and potential for integration into stretchable electronics further highlight its potential in next-generation soft energy storage systems.

Figure 6. Electrochemical performance of the supercapacitor using the PVA-CDs-SP hydrogel as a quasi-solid-state electrolyte. (a) Schematic illustration of the assembled

supercapacitor configuration. (b) CV curves at different scan rates, showing ideal capacitive behavior. (c) GCD curves at various current densities. (d) Capacitance retention at different current densities, demonstrating rate capability. (e) Cycling stability of the device over 8000 charge-discharge cycles.

8. SEM images of PVA-CDs reveal significant black spots ($\sim 2 \mu\text{m}$) defect. These require further analysis and discussion to clarify their origin and implications.

A: Thank you for your constructive feedback. We appreciate your observation regarding the black spots ($\sim 2 \mu\text{m}$) observed in the SEM images of PVA-CDs. These spots are not defects but are a result of the sublimation of water during the freeze-drying process of the hydrogel. As the water evaporates, it leaves behind these characteristic marks, which are commonly observed in freeze-dried hydrogels. We have now included a more detailed explanation of this phenomenon in the manuscript, clarifying that these black spots are artifacts from the freeze-drying process rather than structural defects. This explanation should help to avoid any confusion regarding the nature of these features. Thank you again for your insightful comment, which has allowed us to improve the clarity of the manuscript.

9. The crystallization domain depicted in the schematic diagram Figure 1e should be contributed from PVA chains. The current schematic may confuse readers regarding the microstructure. A revised and clearer diagram is essential.

A: Thank you for your insightful comment. We understand your concern regarding the schematic diagram in Figure 1e. We agree that the crystallization domain should be attributed to the PVA chains, and the current diagram may indeed cause confusion regarding the microstructure. In response to your feedback, we have revised the schematic to more clearly indicate that the crystallization domain arises from the PVA chains. The updated diagram now more accurately reflects the microstructure of the

material, making it easier for readers to understand the crystallization behavior. We believe this revision improves the clarity of the manuscript and enhances the overall presentation. Thank you again for your valuable suggestion.

Fig. 1 | Preparation of CDs and PVA-CDs-SP hydrogel. **a** Schematic diagram of the preparation of CDs. **b** Transmission electron microscopy (TEM) images of CDs and its particle size distribution (insert). **c** Raman spectrum of CDs. **d** X-ray diffraction (XRD) spectrum of CDs. **e** Schematic diagram of the preparation of PVA-CDs-SP hydrogel. **f**

Scanning electron microscopy (SEM) images of the surfaces (up) and cross-sections (down) of PVA, PVA-CDs and PVA-CDs-SP hydrogel.

I hope these comments help guide future revisions and improve the quality of the work.

A: We sincerely thank you for your thorough review and valuable comments on our manuscript. Your insightful suggestions have greatly helped us improve the quality of our research and have made the manuscript more rigorous and comprehensive. We deeply appreciate your input and have carefully revised the manuscript based on your feedback to enhance its scientific merit and readability. Once again, thank you for your time and contribution — your feedback has been instrumental to the improvement of our work.

Reviewer #3:

• However, the tensile strength reported in this work was not a significant increase compared with the reported papers. Some reported hydrogels also showed tensile strength over 100 MPa besides the cited references in this work, such as *J. Mater. Chem. B*, 2018,6, 8105. Additionally, the authors did not give the water content of the hydrogels with ultra-high strength, and the comparison of hydrogel strength could not be separated with its water content. Thus, the novelty of in strength was not so high.

A: Thank you for your valuable and insightful comment. We fully agree that the comparison of hydrogel strength should take water content into careful consideration, as it significantly influences mechanical performance. In the revised manuscript, we have added data on the water content of the hydrogels under different progressive stretching conditions. Even under the lowest water content, the hydrogel still retains more than 25% water. We acknowledge that this is relatively low for conventional hydrogels, and indeed, to achieve the reported high mechanical strength, a certain compromise in water content was necessary. Similar trade-offs have been observed in other reported high-strength hydrogels, such as the one in *Adv. Mater.* 2024, 36, 2313845, where the water content is approximately 20%. We appreciate your helpful suggestion, which has improved the completeness and clarity of our manuscript.

Supplementary Figure 14. (a) Stress–strain curves of PVA-CDs-SP hydrogels at different elongations, and (b) water content of PVA-CDs-SP hydrogels at different

elongations.

Additionally, to illustrate the changes in the mechanical properties of the hydrogel during the progressive stretching process, PVA-CDs-SP hydrogels under different progressive stretches were tested, as shown in Supplementary Fig. 14a. As the progressive stretching proceeded, the mechanical properties of the PVA-CDs-SP hydrogel gradually increased, but this was accompanied by a decrease in the elongation at break, which is due to the fact that after extensive progressive stretching, the PVA-CDs-SP hydrogel gradually approaches an oriented state. Furthermore, as the progressive stretching continued, the water content of the PVA-CDs-SP hydrogel gradually decreased (Supplementary Fig. 14b), because during the stretching process, the PVA chains tended to align, causing the internal water content to decrease.

- Moreover, the discussion in toughness is not probable, fracture toughness of the hydrogel is usually defined in reference to the initiation and growth of a pre-existing crack under prescribed loading (refer to Soft Matter, 2016, 12, 8069). The “toughness” showed in this paper is just the extension work of the hydrogel sample. This is a common problem with some reported papers.

A: Thank you for your reminder. This was due to our insufficient consideration. We have added a new discussion on the fracture energy of hydrogels.

To further confirm the ability of PVA-CDs-SP hydrogel to retard crack propagation, the crack sensitivity of the PVA-CDs-SP hydrogel was also investigated. Interestingly, we observed that with the increase in pre-stretching, the arrangement of polymer chains in the hydrogel became significantly enhanced, leading to a higher degree of orientation. However, this also resulted in a significant reduction in the elongation ability of the notched hydrogel (Supplementary Fig. 15a). Therefore, although the orientation was improved, the measured fracture energy did not increase as expected. This suggests that,

while the oriented structure is beneficial for tensile strength and stiffness, in a highly ordered state, it may simultaneously limit energy dissipation around the crack tip (Supplementary Fig. 15b). To investigate the mechanical anisotropy of the material, the tensile strength of the hydrogel in the vertical direction was measured (Supplementary Fig. 16). Clearly, in the vertical direction, the material's fracture strength and fracture elongation in the oriented direction are significantly higher than the corresponding values in the vertical direction, which is a result of the molecular alignment in the oriented direction.

Supplementary Figure 15. (a) Stress–strain curves of PVA-CDs-SP notched hydrogels at different elongations. (b) Fracture energy of PVA-CDs-SP notched hydrogels at different elongations.

Supplementary Figure 16. Stress–strain curves of PVA-CDs-SP hydrogels in the vertical direction (R direction) at different elongations.

- At the same time, stable underwater sensor has also been widely reported, but this work did not demonstrate the superiority of high strength of the hydrogel in the underwater sensors.

A: Thank you for your thoughtful comments and suggestions regarding the application of the hydrogel. You are correct that while these properties are impressive, they may not be strictly necessary for underwater sensing, which was our initially stated application. In response to your suggestion, we have now explored and demonstrated a more specific application of the hydrogel by integrating it into ultra-strong aqueous supercapacitors.

Electrochemical Behavior in Supercapacitors

To further explore the practical applications of the PVA-CDs-SP hydrogel, we assembled a quasi-solid-state supercapacitor by integrating the hydrogel as the electrolyte and separator, while using activated carbon (AC) as the electrode material

(Fig. 6a). As shown in Fig. 6b, the cyclic voltammetry (CV) curves at various scan rates exhibit nearly rectangular shapes with minimal distortion even at high scan rates, indicating excellent electrochemical reversibility and ideal capacitive behavior.

The galvanostatic charge-discharge (GCD) curves at different current densities (Fig. 6c) display highly symmetrical isosceles triangle shapes, characteristic of electric double-layer capacitors. The specific capacitance, calculated from the GCD profiles, reaches 91.5 F g^{-1} at a current density of 0.2 A g^{-1} . Even when the current density was increased by 15 times, the device retained 59.8% of its capacitance (Fig. 6d), demonstrating good rate capability. Furthermore, the supercapacitor based on the PVA-CDs-SP hydrogel exhibited excellent long-term cycling stability, maintaining nearly 100% capacitance retention after 8000 charge-discharge cycles (Fig. 6e). These results clearly demonstrate that the PVA-CDs-SP hydrogel not only offers exceptional mechanical robustness but also serves as an efficient quasi-solid-state electrolyte for high-performance supercapacitors. In addition, the outstanding ionic conductivity, water-retention ability, and structural stability of the hydrogel contribute to maintaining a stable ionic environment at the electrode-electrolyte interface, which is crucial for reliable energy storage performance under flexible or wearable conditions. Its adaptability to deformation and potential for integration into stretchable electronics further highlight its potential in next-generation soft energy storage systems.

Figure 6. Electrochemical performance of the supercapacitor using the PVA-CDs-SP hydrogel as a quasi-solid-state electrolyte. (a) Schematic illustration of the assembled supercapacitor configuration. (b) CV curves at different scan rates, showing ideal capacitive behavior. (c) GCD curves at various current densities. (d) Capacitance retention at different current densities, demonstrating rate capability. (e) Cycling stability of the device over 8000 charge-discharge cycles.

• Furthermore, this manuscript was not well organized, some description was not clear, such as line 304-312, the “carbonization” is not clear, “the toughness of PVA-CDs-SP hydrogel prepared at 160 °C reached ...”, how did the hydrogel be prepared at 160 °C?

A: Thank you for your careful reading and constructive feedback regarding the clarity and organization of the manuscript. We sincerely apologize for the confusion caused by the unclear description in lines 304–312. In the revised version, we have reorganized and clarified this section. Specifically, we have rephrased the sentence to explain that “160 °C” refers to the temperature used during the thermal treatment of citric acid to synthesize carbon dots (CDs), not the temperature used for hydrogel preparation. The

CDs were obtained via a carbonization process at 160 °C and then incorporated into the PVA hydrogel matrix. The hydrogel itself was prepared under mild conditions, as described in the Methods section. We have revised the wording to explicitly distinguish between the synthesis temperature of CDs and the hydrogel fabrication process to ensure readers are not misled. Thank you again for pointing this out — your feedback has helped us improve the clarity and structure of the manuscript.

The results indicate that the optimal temperature for preparing CDs is 160°C. At this temperature, the tensile strength of the PVA-CDs-SP hydrogel prepared from the carbon dots reaches 154.5 MPa, which is significantly higher than that of hydrogels prepared from carbon dots burned at other temperatures (Fig. 3a). In addition, the toughness of the PVA-CDs-SP hydrogel prepared from the carbon dots burned at 160°C reaches its maximum value of $210.3 \pm 21.9 \text{ MJ m}^{-3}$ (Fig. 3b).

- Some data in the figures was repeated, such as data in Figure 2a and b, etc. It is not necessary to show the figures in this way.

A: Thank you for your careful observation and helpful comment. We acknowledge that some data, such as those in Figure 2a and 2b, were partially repeated. Our original intention was to present the same data sets from different perspectives to emphasize specific features. However, we understand your concern that such repetition may appear redundant. In response, we have revised the figures to avoid unnecessary duplication and have reorganized the presentation to make it more concise and focused, while preserving the key information. We believe this adjustment improves the clarity and overall flow of the manuscript.

We sincerely appreciate your suggestion, which helped us enhance the visual and logical quality of our work.

Fig. 2 | Mechanical properties of hydrogels and toughening tuning via the addition of CDs. **a** Stress-strain curves of PVA-CDs hydrogels at different concentrations of CDs. **b** Comparison of the mechanical properties of PVA, PVA-CA and PVA-CDs hydrogels. **c** FT-IR spectra of PVA, PVA-CA and PVA-CDs hydrogels. **d** Raman spectra of the O-H stretching vibrational peaks of PVA, PVA-CA and PVA-CDs hydrogels. **e** The low-field nuclear magnetic resonance (L-NMR) curves of PVA, PVA-CA and PVA-CDs hydrogels. **f** The corresponding T₂ and τ_c values of PVA, PVA-CA and PVA-CDs hydrogels. **g** DSC analysis of PVA, PVA-CA and PVA-CDs hydrogels. **h** Enthalpy changes of melting and crystallinity of PVA, PVA-CA and PVA-CDs hydrogels. **i** SAXS spectra of PVA, PVA-CA and PVA-CDs hydrogels. **j** The interlayer spacing of PVA, PVA-CA, and PVA-CDs hydrogels.

- Some experimental details were not given, such as the sample preparation process of each test, the photos of different hydrogels used in this paper, the size of the hydrogels used in the mechanical tests, as well as the details of SEM test.

A: Thank you for your valuable comment. We acknowledge that some experimental details were missing in the original submission. In response, we have carefully revised the manuscript and supplementary information to include the necessary details regarding sample preparation, hydrogel dimensions for mechanical tests and SEM test conditions.

We appreciate your feedback, which has helped us improve the completeness and clarity of our work.

The dimensions of the PVA, PVA-CA, and PVA-CDS hydrogel samples are 10 mm in width, 2 mm in thickness, and 40 mm in length.

Preparation of PVA-CA_{10%} hydrogel

PVA was dissolved in a 10% citric acid (CA) solution and stirred at 90 °C for 3 hours to obtain a PVA-CA_{10%} solution. The resulting mixture was poured into a silicone mold, frozen at -20 °C for 12 hours, and then thawed at room temperature for 2 hours. This freeze-thaw cycle was repeated three times to obtain the PVA-CA_{10%} hydrogel. Unless otherwise specified in the main text, PVA-CA refers to the PVA_{20%}-CA_{10%} composition.

Preparation of PVA_n-CDs_m hydrogel

PVA was dissolved in the CDs solution and stirred at 90 °C for 3 hours to obtain PVA_n-CDs_m solutions with different concentrations. The homogeneous solution was then poured into silica molds, frozen at -20 °C for 12 hours, thawed at room temperature for

2 hours, and subjected to three freeze-thaw cycles to obtain the PVA_n-CDs_m hydrogel.

Here, n represents the concentration of PVA in the PVA-CDs hydrogel, m represents the concentration of the CDs solution.

Unless otherwise specified in the main text, PVA-CDs refers to the PVA_{20%}-CDs_{10%} composition.

Preparation of PVA-CDs-SP_x hydrogel

PVA-CDs-S hydrogel was then gradually stretched to 100%-600% of its original length and subsequently soaked in a sodium citrate solution for another 12 h to obtain PVA-CDs-SP_x hydrogel. Here, X represents the PVA-CDs-S hydrogel stretched to different strain levels. The hydrogel codes and the weight fractions of each component in the hydrogels were summarized in **Supplementary Table 1**. Different CDs prepared with malic acid, tannic acid and eucalyptus bark to verify the universality. All parameters were consistent with those used in the preparation of PVA-CDs-SP hydrogels.

Scanning electronic microscopy (SEM) Analysis

The hydrogel was freeze-dried, and then its morphology was observed using scanning electron microscopy (SEM, ZEISS GeminiSEM 300, Germany). The freeze-dried hydrogel was directly adhered to conductive adhesive and coated for 45 seconds at 10 mA using a Quorum SC7620 sputter coater.

Polarized light microscope testing

The crystalline morphology of the sample was observed using a polarized light optical

microscope (OLYMPUS GX71).

Supplementary Table 1. The compositions of the PVA, PVA-CA and PVA-CDs hydrogels.

Sample	PVA (g)	CA (g)	CDs (g)	Water (g)
PVA _{10%}	10			90
PVA _{20%} -CA _{10%} (PVA-CA)	20	8		72
PVA _{10%} -CA _{10%}	10		9	81
PVA _{15%} -CDs _{10%}	15		8.5	76.5
PVA _{20%} -CDs _{10%} (PVA-CDs)	20		8	72
PVA _{25%} -CDs _{10%}	25		7.5	67.5
PVA _{20%} -CDs _{1%}	20		0.8	79.2
PVA _{20%} -CDs _{5%}	20		4	76
PVA _{20%} -CDs _{15%}	20		12	68

Response: Once again, we would like to thank the reviewers for their careful reading and providing insightful feedback that helped us to further improve the manuscript. We thank the reviewers for recognizing the highlights of our work. We have carefully answered all your concerns and technical questions, and we believe that the quality and scientific content of our work has improved significantly. We hope that our revisions will address the reviewers' concerns and convince them to support our manuscript for publication in Nature Communications.

Here, we will endeavor to demonstrate why our work is sufficiently novel and impactful to be relevant to the wider readership of Nature Communications.